# A mapping framework of competition–cooperation QTLs that drive community dynamics

Libo Jiang[1], Xiaoqing He [1], Yi Jin[1], Meixia Ye[1], Mengmeng Sang[1], Nan Chen[1], Jing Zhu[1], Zuoran Zhang[1], Jinting Li[1] & Rongling Wu [1,2]

Genes have been thought to affect community ecology and evolution, but their identification at the whole-genome level is challenging. Here, we develop a conceptual framework for the genome-wide mapping of quantitative trait loci (QTLs) that govern interspecific competition and cooperation. This framework integrates the community ecology theory into systems mapping, a statistical model for mapping complex traits as a dynamic system. It can characterize not only how QTLs of one species affect its own phenotype directly, but also how QTLs from this species affect the phenotype of its interacting species indirectly and how QTLs from different species interact epistatically to shape community behavior. We validated the utility of the new mapping framework experimentally by culturing and comparing two bacterial species, *Escherichia coli* and *Staphylococcus aureus*, in socialized and socially isolated environments, identifying several QTLs from each species that may act as key drivers of microbial community structure and function.

[1] Center for Computational Biology, College of Biological Sciences and Technology, Beijing Forestry University, 100083 Beijing, China. [2] Center for Statistical Genetics, The Pennsylvania State University, Hershey, PA 17033, USA. These authors contributed equally: Libo Jiang, Xiaoqing He, Yi Jin, Meixia Ye. Correspondence and requests for materials should be addressed to R.W. (email: rwu@bjfu.edu.cn)

N o species lives in isolation. The pattern of how one species interact with others affects the behavior and process of ecological communities[1]. A mounting body of evidence has suggested that interspecies interactions determine how communities respond to environmental perturbations, including climate change[2,3]. It has been increasingly recognized that genes from each coexisting species play a pivotal role in shaping the internal workings of communities[4–6], but how to identify these genes at the genome-wide level has been an unsolved issue. Many studies using a simple experimental design have been able to characterize single genes or pathways that contribute to ecological interactions in a community[7–9], but they have proven to be difficult for charting the comprehensive genetic architecture of how different species interact and communicate. As of today, we still cannot build a precise genotype–phenotype map for interspecies interactions in a population, community, or ecosystem, despite the increasing availability of genetic data by high-throughput genotyping and sequencing techniques[10].

Given its capacity to genome-wide search for quantitative trait loci (QTLs) that control complex phenotypes, genetic or association mapping has been widely used as an approach for studying complex genetic questions[11,12]. By integrating the mathematical function of trait formation, a new approach, named functional mapping, has been developed to cope with the developmental feature of complex traits[13,14]. More recently, functional mapping has been upgraded to the level of systems mapping by regarding a complex trait as a system composed of interactive components[15,16]. The characteristic of systems mapping lies in its seamless implementation of a group of ordinary differential equations (ODEs) that can not only characterize the dynamic change of one component, but also discern how one component interacts and coordinates with its partners in a complex system. Thus, by dissolving the phenotype into its underlying components based on morphogenetic, physiological, and anatomic principles[15], systems mapping can map and identify specific QTLs that govern the interconnections of different components, gleaning new insight into the mechanistic basis of trait formation and progression.

To take conceptual advantage of systems mapping, we equip it with the community ecology theory to put forward a new mapping framework. A community or biocoenosis is defined as a dynamic assemblage of populations composed of two or more distinct species occupying the same geographical area[17]. Community ecology examines how the community works as a functional unit through the emergent interactions and coordination of its components[18,19]. It divides interspecies interactions into different types based on whether a species chooses to compete or cooperate with other species, and provides a biological interpretation of how each type of interaction gives rise to the dynamic change and evolution of communities[19–22]. By mathematically modeling the pattern and strength of ecological interactions among different species, the new mapping framework can characterize, quantify, and visualize how QTLs mediate interspecies competition or cooperation. Beyond a reductionist approach for mapping the genotype–phenotype relationship of a single species, our new mapping framework links genotype combinations between interacting species to their phenotypes and further capture how QTLs from one species affect the phenotype of its coexisting species and how QTLs from different genomes interact epistatically to influence the phenotypes of multiple species. The encapsulation of these previously omitted indirect genetic effects and genome–genome epistatic interaction effects would gain new insight into the global genetic architecture of community dynamics.

We carry out an experiment of microbial competition to validate the practical application of the new mapping framework.

A set of strains from *Escherichia coli* and *Staphylococcus aureus* are randomly paired to form 45 independent interspecific pairs, i.e., no single strain from one species was paired with multiple strains from the second species. We create a socialized environment by co-culturing each pair in a flask. To investigate how the bacteria respond to ecological interactions, we also culture each strain in a socially isolated flask. By analyzing the abundance data of paired bacterial strains, the new mapping framework identifies several key QTLs from each species that govern competition and collaboration in microbial communities through their direct, indirect, and epistatic genetic effects. Our framework provides a tool to showcase and infer community structure, organization, and function based on genetic blueprints of the interacting species.

## Results

**Fitting bacterial growth curves**. The behavior of microorganisms under a particular condition follows some biological rule, for example, microbial growth is characterized by different phases, lag, exponential, and stationary[23,24]. In the lag phase, the specific growth rate starts from zero and then accelerates to a maximal value in the exponential phase. When growth reaches the stationary phase, the rate decreases and finally approaches zero. The three phases are each specified by a distinct parameter: the exponential phase by the maximum-specific growth rate ($r$, the tangent in the inflection point), the lag phase by the lag time ($\lambda$, the $x$-axis intercept of this tangent), and the stationary phase by the asymptote ($A$, the maximal value reached). A number of growth equations have been derived to capture these phases, among which three representatives are Gompertz, logistic, and Richards, expressed as

$$N(t) = A\exp\left\{-\exp\left[\frac{r \cdot e}{A}(\lambda - t) + 1\right]\right\} \quad \text{(Gompertz equation)}$$

(1a)

$$N(t) = \frac{A}{1 + \exp\left[\frac{4r}{A}(\lambda - t) + 2\right]} \quad \text{(Logistic equation)} \quad (1b)$$

$$N(t) = A\left\{1 + v\exp(1 + v)\exp\left[\frac{r}{A}\left(1 + v\left(1 + \frac{1}{v}\right)\right)(\lambda - t)\right]\right\}^{-\frac{1}{v}}$$

(Richards equation)

(1c)

where $N(t)$ is the abundance of a microbe at time $t$ and the Richards equation includes an additional parameter ($v$) that describes the shape of a curve.

By assuming the four-parameter Richards equation as one that can exactly predict microbial growth in monoculture, we implemented a statistical procedure to test and validate whether any of the three-parameter Gompertz and logistic equations can sufficiently describe the data. Results from the $F$ test for model comparison suggest that the Richards equation provides an optimal fitness to time-dependent abundance data for both *E. coli* and *S. aureus* in monoculture (Fig. 1). Supplementary Table 1 gives the estimates of growth parameters by each growth equation for each species. Using Sun et al.'s heterochronic formula[25], we divided the Richard curve of microbial growth into lag, exponential, and stationary phases, which span 0–0.54, 0.54–9.14, and 9.14–36 h for *E. coli* and 0–1.21, 1.21–13.49, and 13.49–36 h for *S. aureus*, respectively, showing that these two species have different forms of growth.

Given their possible interactions between coexisting strains, we introduced a Lotka–Volterra (LV) ordinary differential equation

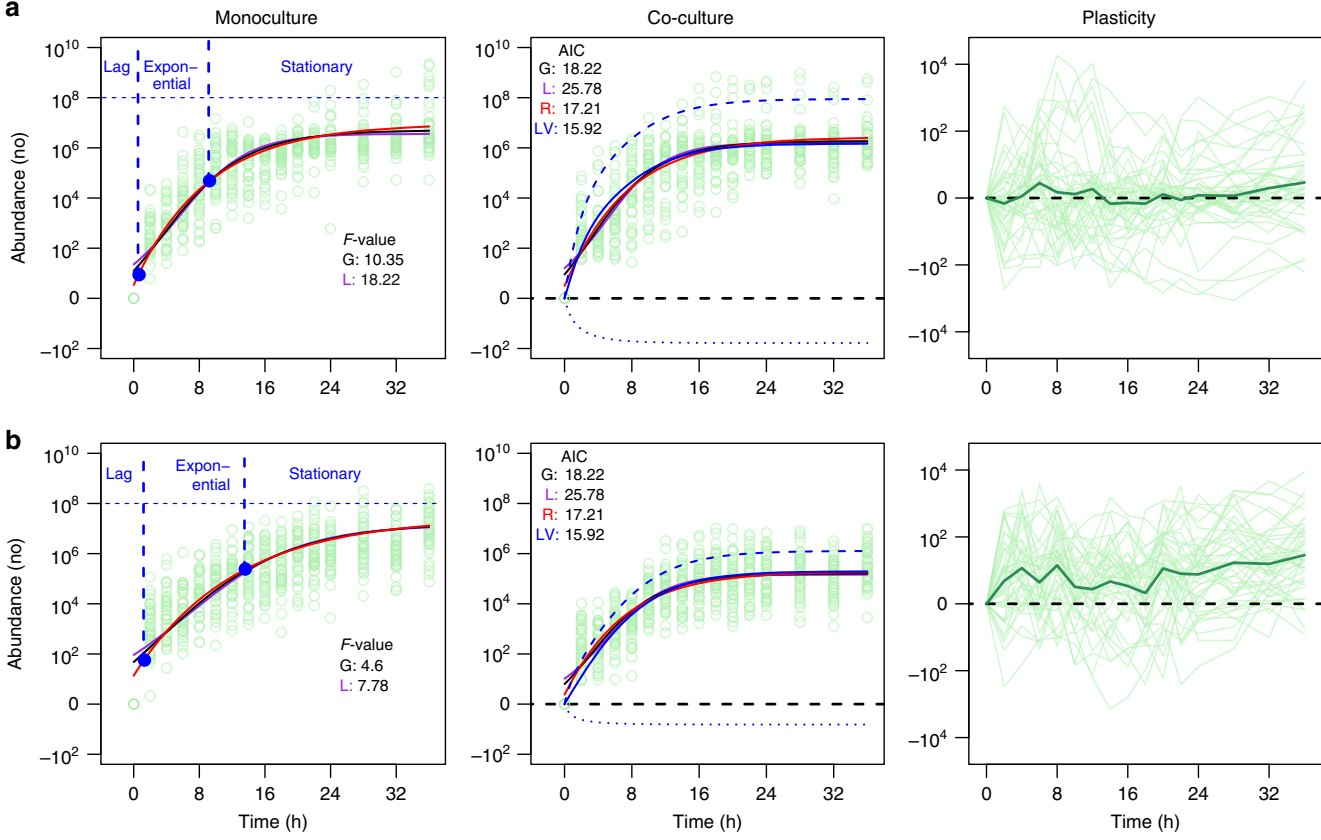

**Fig. 1** Trajectories of microbial growth and its interaction-induced phenotypic plasticity for *E. coli* (**a**) and *S. aureus* (**b**). Microbial abundance of individual strains from each species (circles) was observed in monoculture and co-culture during the first 36 h after culture. In monoculture, the *F* test of three-parameter Gompertz (G, green solid line) and logistic equations (L, red solid line) against the four-parameter Richards equation (R, blue solid line) suggests that the latter gives a better goodness-of-fit to mean growth trajectories than the former two equations. Three distinct phases, lag, exponential, and stationary, on the Richards curve are indicated. In co-culture, the G, L, and R equations were compared with the Lokta–Volterra (LV) ordinary differential equations, suggesting that the LV is the most parsimonious according to the AIC values. The LV curve of each species in co-culture was partitioned into its underlying independent (broke line) and interactive growth components (dot line). Negative interactive growth for both species shows an antagonistic relationship held between *E. coli* (**a**) and *S. aureus*. Phenotypic plasticity curves were calculated as the differences of bacterial growth in monoculture from that in co-culture at different time points. Mean plasticity curves are indicated by thick lines

group[24] to jointly fit the microbial growth of two species in co-culture. Let $N_e$ and $N_s$ denote the abundances of two coexisting strains from *E. coli* and *S. aureus*, respectively. The LV equations are expressed as

$$\begin{cases} \dot{N}_e = r_e N_e \left(1 - \frac{N_e + \alpha_{e|s} N_s}{K_e}\right) \\ \dot{N}_s = r_s N_s \left(1 - \frac{N_s + \alpha_{s|e} N_e}{K_s}\right) \end{cases} \quad (2)$$

where $r_e$ and $r_s$ are the Malthusian growth rates of *E. coli* and *S. aureus* strains, respectively; $K_e$ and $K_e$ are an intrinsic-carrying capacity of two different species; and $\alpha_{e|s}$ and $\alpha_{s|e}$ are dimensionless parameters that model how one species affects the other through competition or cooperation in co-culture. We fit the growth data of the two bacterial species in co-culture using Gompertz, logistic, Richards, and LV equations (Supplementary Table 1) and further chose one that best fit the data based on the AIC information criterion. The result suggests that the LV equations outperform the Gompertz, logistic, and Richards equations in fitting growth trajectories of both species in co-culture (Fig. 1). A random relationship between the predicted values and the residuals across individual interspecific pairs (Supplementary Fig. 1) indicates that the LV model possesses reasonably good statistical behavior in our co-culture data fitting.

To explore the consequence of interspecies interactions, we compared the difference of growth trajectories between the same species in monoculture and co-culture, i.e., phenotypic plasticity induced by microbial coexistence[22]. On average, the growth trajectories of *E. coli* are slightly responsive to its conspecific (Fig. 1a), whereas *S. aureus* displays measurable phenotypic plasticity, with a greater growth rate in monoculture than in co-culture (Fig. 1b). Considerable variation was detected in growth trajectory among different strains from each species in each culture (Fig. 1), implicating the possible existence of QTLs that regulate microbial growth. For both species, a great variability exists in how a strain responds to ecological coexistence (Fig. 1). Some strains grow better in co-culture than in monoculture, while others exhibit an opposite pattern of response. Furthermore, the sign and magnitude of phenotypic plasticity may vary over the time of culture.

**Mapping microbial abundance by a static approach.** We used two approaches, static and dynamic, to map growth QTLs in two cultures. The static mapping approach is to associate each genome with microbial abundances of strains measured at individual time points, whereas the dynamic mapping approach attempts to regress growth trajectories of strains on their marker genotypes through mathematical equations. After adjusting the phenotypic

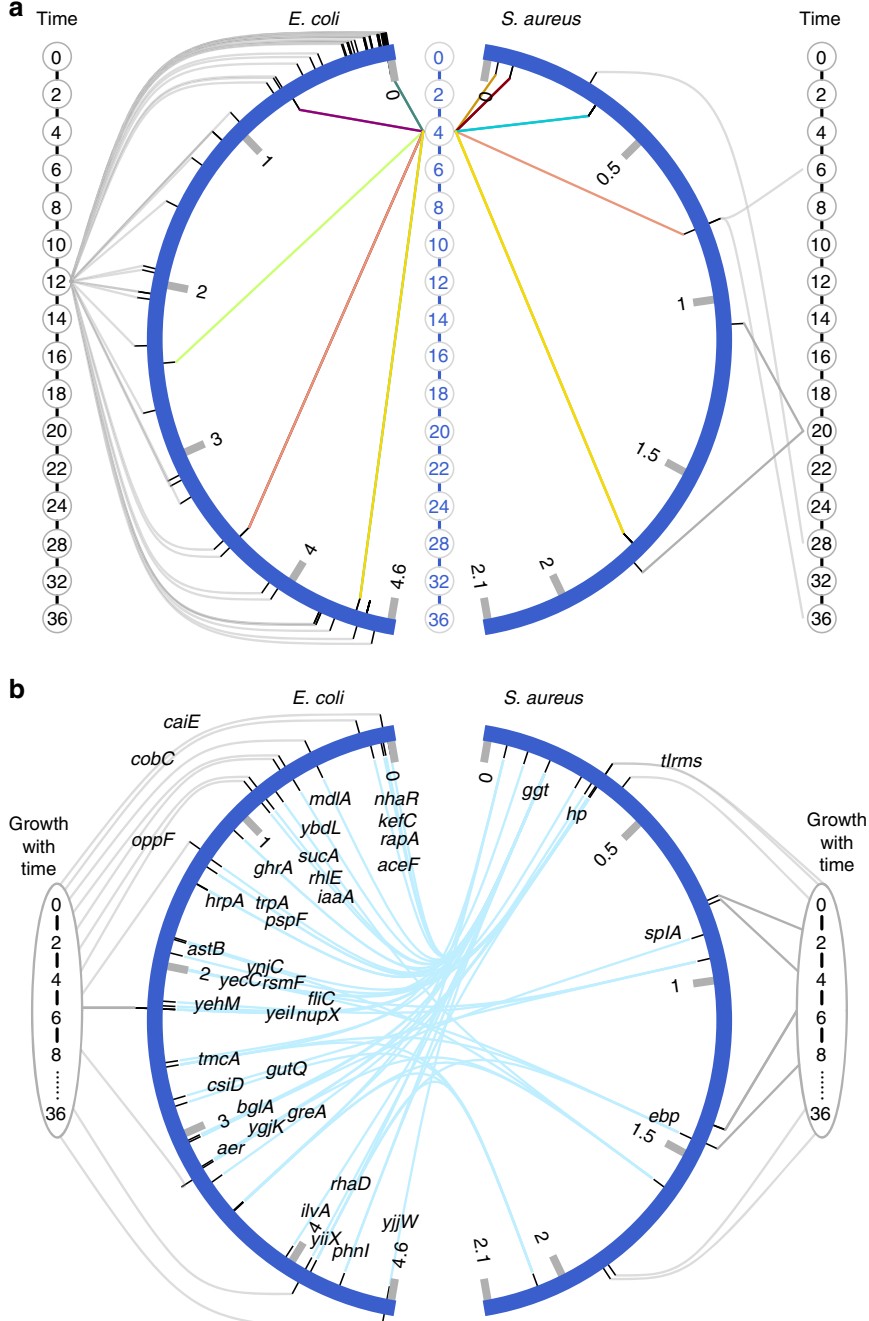

**Fig. 2** Identification of significant QTLs associated with the microbial growth of *E. coli* and *S. aureus* in monoculture and co-culture. **a** QTLs for time-dependent abundance detected by a static model. Outside and inside the genomes of each species (semi-circle) are the aligned time points, 0, 2, 4,..., 36 (in hour), at which abundance data were collected in monoculture and co-culture, respectively. Thin lines indicate significant associations between QTLs and microbial abundance at a particular time point. A pair of QTLs from *E. coli* and *S. aureus* that jointly affect the two species' abundance in co-culture are shown by two lines in the same color. **b** QTLs for microbial growth trajectories over time 0, 2, 4,...,36 detected by a dynamic model. Outside and inside the genomes of each species (semi-circle) are QTLs or their underlying candidate genes detected in monoculture and co-culture, respectively. A pair of QTLs from different species to affect microbial growth are connected by thin lines. A genome-wide significance threshold for QTL detection at the 5% level was determined by 1000 permutation tests

data for population structure aimed to avoid the detection of spurious associations, both approaches identified different sets of QTLs for each species, depending on where it was grown (Fig. 2). In monoculture, the static approach found a number of significant QTLs (63) in *E. coli* distributed throughout the genome, but all these QTLs are only associated with abundance at time 12 h of the early stationary growth phase (Fig. 2a). Under the same condition, only five QTLs were observed for *S. aureus*,

with one affecting the exponential phase and four affecting the stationary phase.

The bacterial abundance of co-culture was mapped by a bivariate model that integrates genetic and phenotypic information of two interactive species *E. coli* and *S. aureus* (Methods). We identified eight QTLs from *E. coli* and six QTLs from *S. aureus* that affect microbial growth through their 17 pairs (Fig. 2a). But these QTL pairs exert a significant influence only on the

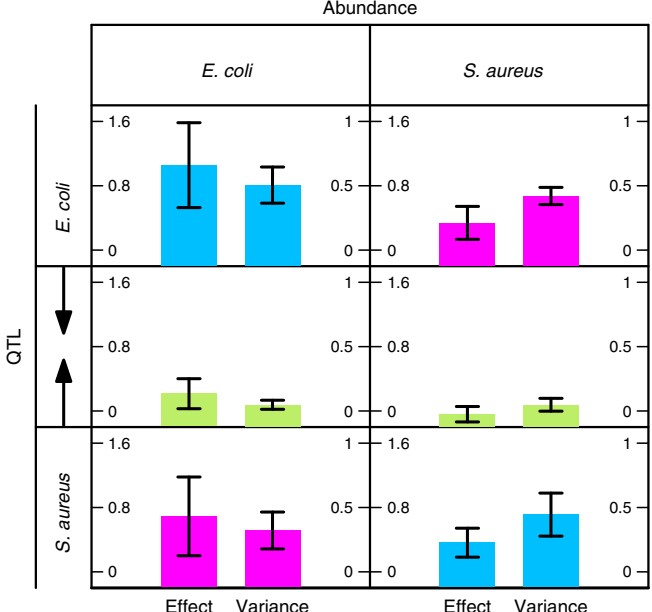

**Fig. 3** Direct, indirect, and genome–genome epistatic genetic effects of QTLs operating in a community. The genotypic value of genome–genome combination at QTL E635206 from *E. coli* and QTL S30869 from *S. aureus* for microbial abundance at 4 h is partitioned into its direct, indirect, and genome–genome epistatic genetic effect components. Upper left: direct effect of E635206 on *E. coli* growth. Lower left: direct effect of S30869 on *S. aureus* growth. Upper right: indirect effect of E635206 on *S. aureus* growth. Lower left: indirect effect of S30869 on *E. coli* growth. Middle: genome–genome epistatic effect of E635206 and S30869 on *E. coli* growth (left) and *S. aureus* growth (right). Within the same plot, left bar is the genetic effect, whereas right bar is the proportion of the total genetic variance at this interspecific QTL pair explained by each effect. The standard errors of each estimated effect are shown

abundance at 4 h, a middle stage of the exponential phase. We further partitioned the genotypic values of four genotype combinations at each QTL pair into direct, indirect, and genome–genome epistatic genetic effect components (Methods). We used a pair of QTLs E635206 from *E. coli* QTL and S30869 from *S. aureus* as an example to explain our discovery (Fig. 3). This QTL pair explains a heritability of 0.035 and 0.022 for the abundance at 4 h in *E. coli* and *S. aureus*, respectively. The two QTLs each trigger an effect not only on the growth of their home species directly ($P < 0.01$), but also on the growth of the opposite species that coexist with their home species indirectly ($P < 0.01$). Also, they affect the growth of each species epistatically across different genomes ($P < 0.01$). Together, indirect and genome–genome epistatic effects at this QTL pair explain 49% and 54% of the total genetic variance for *E. coli* and *S. aureus*, respectively. We calculated the total heritability of 4 h abundance explained by all significant 17 QTL pairs (Fig. 2a), which is 0.39 for *E. coli* and 0.25 for *S. aureus*. Indirect and genome–genome epistatic effects of all QTL pairs together account for 59% and 48% of the total heritability of these two species, respectively. Approximately one half of the heritability was contributed by these two interspecific interaction-induced genetic effects.

**Modeling the genotype–phenotype relationship by a dynamic approach**. As two dynamic approaches for QTL mapping, functional mapping, and systems mapping were used to map microbial growth in monoculture and co-culture, respectively. By implementing the best-fit Richards equations, functional mapping

identified 24 significant SNPs associated with time-dependent abundance for both species in monoculture (Fig. 2b). Results from the functional annotation of genes using NCBI's GenBank® sequence database show that a large portion of QTLs detected by functional mapping residue in genome positions of candidate genes, validating the usefulness and statistical power of this approach, as well demonstrated in previous studies[14].

Systems mapping implemented by the LV equations can identify QTLs that control the interactive pattern of two species in co-culture (Methods). We integrated this approach and community ecology to produce a new mapping framework by which to quantify and interpret the pattern of interspecies interactions and characterize and map specific QTLs that regulate each of these interactions. Any species in co-culture may choose to cooperate or compete with its conspecific, depending on at which level the common resource can be shared for their respective growth[19]. According to the community ecology theory, the strategy with which these two species interact with each other can be formulated by a strategy matrix

$$
\begin{array}{c}
 & \textit{S. aureus} \\
\textit{E. coli} \;
\begin{array}{c} - \\ 0 \\ + \end{array}
\left(
\begin{array}{ccc}
\text{Antagonism} & \text{Amensalism} & \text{Parasitism} \\
\text{Amensalism} & \text{Independence} & \text{Commensalism} \\
\text{Parasitism} & \text{Commensalism} & \text{Mutualism}
\end{array}
\right)
\end{array}
\tag{3}
$$

Different strategies used by each species lead to six distinct interaction types: mutualism by which two species benefit from each other, independence by which any one species does not depend on or affects the other, antagonism by which two species are in conflict of one another, commensalism by which one species benefits its conspecific whereas the latter does not affect the former, predation/parasitism by which one species helps the other but the latter is harmful to the former, and amensalism by which one species hurts the other while the latter does not affect the former.

These competition or cooperation relationships as well as their strengths can be captured by the LV equations. We split the LV equations into two different parts that describe microbial abundance differently, expressed as

$$
\begin{cases}
\dot{N}_e = r_e N_e \left(1 - \frac{N_e}{K_e}\right) + r_e N_e \left(\frac{-\alpha_{e|s} N_s}{K_e}\right) \equiv \dot{M}_e + \dot{N}_{e|s} \\
\dot{N}_s = r_s N_s \left(1 - \frac{N_s}{K_s}\right) + r_s N_s \left(\frac{-\alpha_{s|e} N_e}{N_s}\right) \equiv \dot{M}_s + \dot{N}_{s|e}
\end{cases}
\tag{4}
$$

where the first part, $\dot{M}_e$ or $\dot{M}_s$, is the independent growth of each species, determined by its intrinsic property, and the second part, $\dot{N}_{e|s}$ or $\dot{N}_{s|e}$, presents the interactive growth of each species, determined by how it interacts with its conspecific. If the interactive growth of a species is positive or negative, this indicates that this species is benefitted or harmed by another species. If there is no interactive growth, it means that the two species are not affected by one another. Thus, by estimating ODE parameters $\Theta = (r_e, K_e, \alpha_{e|s}; r_s, K_s, \alpha_{s|e})$, the LV equations can not only specify the dynamic pattern of abundance for each species, but also characterize the interactive pattern of two species in co-culture. On average, *E. coli* and *S. aureus* when cultured together in the same environment were detected to hold an antagonistic relationship since the interactive growth of each species is negative (Fig. 1).

The new mapping framework developed through a unifying likelihood of two interactive species (Methods) can characterize how QTLs determine interspecies competition and cooperation in

a community. As such, it is named the competition–cooperation mapping (CoCoM) model. We used CoCoM to pairwise scan markers from different species throughout their genomes, obtaining 54 significant combinations of QTLs comprising of 41 SNPs from *E. coli* and 12 SNPs from *S. aureus* (Fig. 2b). This number detected by dynamic CoCoM is remarkably larger than the number of QTLs by static CoCoM (Fig. 2a), possibly suggesting the increasing power of QTL detection by the former. The functional annotation of genes from NCBI's database shows that almost all QTLs detected residue within candidate genes of known biological functions, ranging from RNA transcription to peptidoglycan metabolism to biotic stress tolerance and evolution (Supplementary Table 2). For *E. coli*, SNPs E1483999 and E1486303 reside in a gene, *hrpA*, that encodes targeted mutagenesis for improving this bacteria's isobutanol tolerance[26]. E19056 is close to gene *nhaR* encoding transcriptional activator NhaR that mediates the osmotic induction of *osmC*(p1), a promoter of the stress-inducible gene *osmC* in *E. coli*[27]. E3328548 is relevant to the *greA* gene that stimulates the mRNA cleavage activity of RNA polymerase, helping to stall or incorporate incorrect nucleotides[28]. For *S. aureus*, SNP S188004 exerts pronounced genetic interactions with many SNPs distributed over the entire *E. coli* genome.

**Comprehending the genetic architecture of community growth**. CoCoM can chart a more complete picture of genetic architecture by revealing previously neglected indirect and genome–genome epistatic genetic effects of QTLs. We explained this power based on a representative QTL pair E4614704 (from *E. coli*) and S188004 (from *S. aureus*). By gene annotation analysis, these two QTLs were detected to be associated with some biological functions. E4614704 resides in *yjjW*, a gene that encodes a homolog of pyruvate-formate lyase activating enzyme PflA[29]. As a key intersection in the network of metabolic pathways, pyruvate can be converted to carbohydrates via gluconeogenesis, fatty acids, or energy through acetyl-CoA, the amino acid alanine, or ethanol, depending on whether the condition is aerobic or anaerobic. All these metabolic processes may play an important role in adapting *E. coli* to microbial coexistence. S188004 is relevant to the gene *ggt*[30]. This gene encodes gamma-glutamyltranspeptidase that regulates the metabolic pathway of glutathione. By converting methylglyoxal to D-lactate, glutathione may form a key pathway for *S. aureus* to react with microbial interactions (Supplementary Fig. 2).

We drew the fitted overall curves of microbial growth for *E. coli* and *S. aureus* in co-culture at four interspecific genotypic combinations of this QTL (Fig. 4a). We divided each curve into its independent ($\dot{M}_e$, $\dot{M}_s$) and interactive growth components ($\dot{N}_{e|s}$, $\dot{N}_{s|e}$). For combination C/C, the overall growth of *E. coli* is considerably smaller than its independent growth because this species is hindered by *S. aureus* to form the negative interactive growth. This is reciprocally true for the overall growth of *S. aureus* (Fig. 4a), suggesting that these two bacteria are in an antagonistic relationship (matrix 2), although the extent to which *S. aureus* confronts *E. coli* is much larger than that to which *E. coli* represses *S. aureus*. A similar antagonistic relationship was also observed for the other three genotypic combinations C/T, T/C, and T/T, but the pattern of antagonistic relationship varies among the four genotypic combinations. All these results suggest that E4614704 and S188004 are antagonistic QTLs that participate in determining and shaping the antagonistic relationship between *E. coli* and *S. aureus* when they are co-cultured in the same medium. This result was confirmed by an additional analysis of the state-space of the combination of abundance between two species (Supplementary Fig. 3).

QTL E4614704 owned by *E. coli* affects directly its own growth; so does S188004 from *S. aureus*. Yet, it is interesting to find that these two QTLs each exhibit pronounced indirect effects and genome–genome epistatic effects, even with magnitudes being larger than those of direct effects (Fig. 4b). It seems that S188004 from *S. aureus* is a more "aggressive" QTL, because its indirect effect on the abundance dynamic of *E. coli* is larger than its direct effect on the abundance of its home species during the exponential phase of growth and also larger than the indirect effect of E4614704 from *E. coli* on the abundance of *S. aureus*. It can be seen that indirect effects and genome–genome epistatic effects together account for a large proportion of the total genetic variance (68–85% for *E. coli* and 57–81% for *S. aureus*) in the abundance trajectories of both species when they are co-cultured. This part of genetic variance cannot be detected by any traditional mapping approach. From genetic effect and variance curves, we can see how an SNP affects the growth of two microbial species over time. At this particular QTL combination, direct, indirect, and epistatic effects on *E. coli* reach a maximum at time 6–10 h, whereas these effects on *S. aureus* are somewhat periodic over time. Overall, the proportions of the direct effects to the total genetic variance in both species tend to decrease with time.

We drew abundance curves of two genotypes at E4614704 for *E. coli* and at S188004 for *S. aureus* in co-culture, respectively, from which to estimate the genetic effect curves of each QTL (Fig. 4c). These effects are marginally significant ($P = 0.095$), but the genetic effects due to differences among four QTL genotype combinations C/C, C/T, T/C, and T/T are highly significant for both species ($P < 0.001$) (Fig. 4a). This suggests that the impact of some QTLs can be activated by their interaction with QTLs from conspecifics in communities. We also compared growth curves of the same genotype from each species between co-culture and monoculture, from which the genetic effect curves of a QTL in each treatment and the allelic sensitivity curves of the same genotype to different treatments can be drawn (Fig. 4c). Dramatic differences in the effect curve between monoculture and co-culture and in the sensitivity curve between two genotypes ($P < 0.01$) suggest the existence of QTL × ecological interactions.

**QTL network**. The abundance of *E. coli* in co-culture is determined jointly by direct effects of its own 41 SNPs, indirect effects of 12 SNPs from *S. aureus*, and interspecific epistatic effects between these SNPs. On contrast, 12 *S. aureus* SNPs and 41 *E. coli* SNPs trigger direct and indirect effects, respectively, on the abundance of *S. aureus* in co-culture. We implemented an ODE-based genetic networking approach to characterize how these QTLs interact with each other in a network to affect microbial abundance through three different types of effects (Fig. 5). The central feature of QTL networks is its capacity to identify hub QTLs that play a pivotal role in the genetic architecture of microbial growth in species coexistence. SNP 11 (E1317124), 16 (E1838489), and 26 (E2594191) from *E. coli* as hub QTLs affect directly its own performance (Fig. 5a), whereas hub *E. coli* QTLs in the network of indirect effects are SNPs 3 (E62059), 33 (E3328548), 35 (E3539840), and 37 (E3956003) (Fig. 5b). Interestingly, SNP 1 (S188004) and 10 (S2076600) from *S. aureus* serve as hub QTLs in both networks of direct and indirect effects, although the structure of the network differs between these two types of effects. Genome–genome epistasis mediates the abundance of *E. coli* and *S. aureus* in a varying but complex network.

**Biological and statistical validation**. One merit of CoCoM is to decompose the overall growth of bacteria in co-culture into its independent and interactive growth components so as to better reveal the quantitative impact on the growth of a species by its

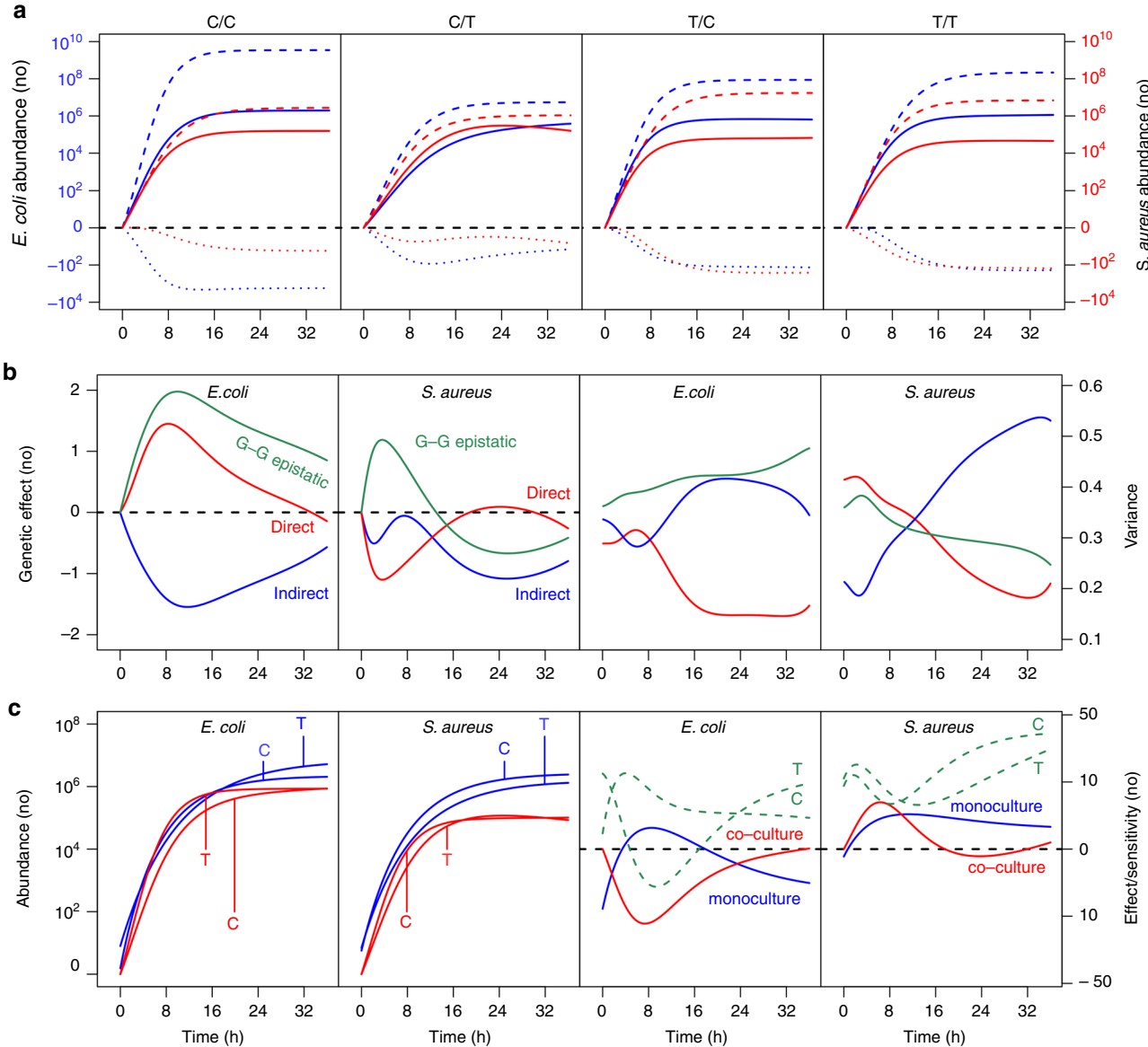

**Fig. 4** Microbial growth curves of genotype combinations across different species in co-culture. **a** QTL E4614704 from *E. coli* and S188004 from *S. aureus* form four genotype combinations C/C, C/T, T/C, and T/T. The overall growth of microbial abundance (solid line) for each combination is decomposed into its independent (broke line) and interactive growth components (dot line) for *E. coli* (blue) and *S. aureus* (red). The interactive curves all are below the zero line, suggesting the negative effect one species exerts on its coexisting species. **b** The genotypic value of each genotype combination is partitioned into its direct (red), indirect (blue), and genome–genome (G–G) epistatic effects (green) on the growth of each species (the first two plots from the left), with the time-varying proportions of the total genetic variance explained by each of these effects (the first two plots from the right). **c** Comparison of microbial growth for the same genotype (C or T) at E4614704 from *E. coli* and the same genotype (C or T) at S188004 from *S. aureus* between co-culture (red) and monoculture (blue) (the first two plots from the left). The genetic effect curves (solid cline) of the same QTL in co-culture (red) and monoculture (blue), as well as the allelic sensitivity curves (broke line) of the same genotype to different culture treatments, were calculated (the two plots from right)

coexisting species and, more importantly, characterize the genetic machineries of this species–species interaction. To demonstrate the biological relevance of the model, we reanalyzed the data of experiment by culturing all strains individually in isolated flasks. The microbial abundance of each species in monoculture was fitted separately for two alternative genotypes at E4614704 for *E. coli* and S188004 for *S. aureus* (Supplementary Fig. 4). These curves that represent microbial growth in isolation were detected to be in a broad agreement with those of independent growth components estimated by CoCoM. This consistency supports the usefulness and accuracy of the model for unveiling the biologically grounded rules underlying the genetic architecture of ecological interactions.

We also conducted computer simulation to validate the statistical properties of CoCoM. The data were simulated by assuming that two species are reared in monoculture and co-culture. The phenotype is determined by a set of QTLs among 1000 simulated markers, plus a residual error following a multi-variate normal distribution. Under monoculture simulation scenario, the genetic component of phenotypic variation was due to direct effects only, whereas the data under co-culture simulation scenario involve all direct, indirect, and interspecific epistatic effects. The data were analyzed reciprocally by traditional univariate functional mapping and the new model. As expected, traditional mapping can effectively detect significant QTLs from the monoculture data under a modest sample size and

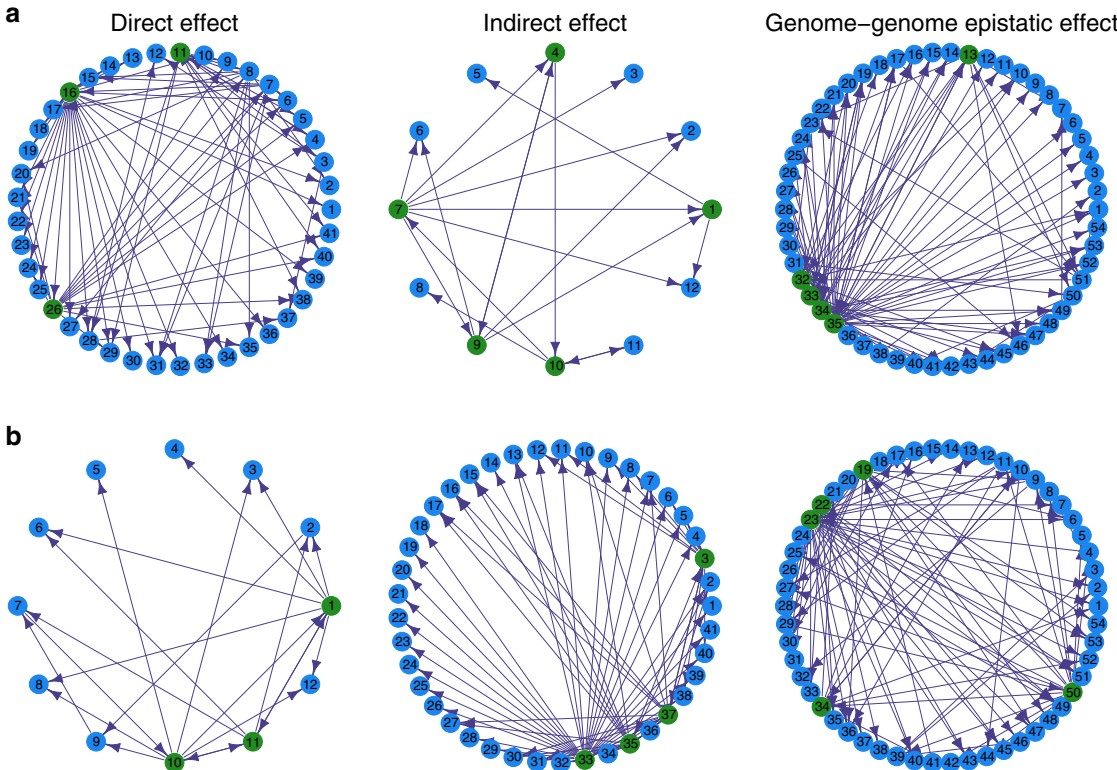

**Fig. 5** QTL networks through different types of genetic effects. **a** Direct effects of 41 *E. coli* QTLs, indirect effects of 12 *S. aureus* QTLs, and inter-genomic epistatic effects of these QTLs on the growth trajectory of *E. coli*. **b** Direct effects of 12 *S. aureus* QTLs, indirect effects of 41 *E. coli* QTLs, and inter-genomic epistatic effects of these QTLs on the growth trajectory of *S. aureus*. Arrows indicate the direction of one QTL/effect to activate another QTL/effect. Names of genes corresponding to each number are given in Supplementary Table 2

heritability, but its power for QTL detection from the co-culture data has reduced sharply to 0.15–0.25 (upper panel, Supplementary Table 3). Similarly, CoCoM shows reasonably good power to detect QTLs hidden in the co-culture data, although it is not proper to analyze the monoculture data. We also drew growth curves using ODE parameters estimated by CCM, in a comparison with the true curves, showing that the true curves are within the confidence interval of the estimated curves (Supplementary Fig. 5). All the above results suggest that CoCoM is not only essential for precisely mapping any QTL that affects phenotypic traits in an interactive community, but also it is statistically robust for the identification of significant QTLs.

As a modeling exploration of community genetics, this pilot study only chose 45 interspecific pairs of strains from two bacterial species. We further examined the statistical properties of the new model to analyze such a small data set. Although its power to detect significant QTLs reduces largely, the estimation of growth curves for each genotype is still within a reasonable estimation interval (lower panel, Supplementary Table 3 and Supplementary Fig. 6). In general, our model provides the reasonable accuracy of parameter estimates when heritability ranges from 0.05 to 0.10 even with a modest number of interspecific pairs, but a sample size of 200 is required for increasing its power.

We performed an additional simulation study to assess the false-positive rate (FPR) of CoCoM. The same scenarios were used to simulate the genetic and phenotypic data under the assumption of no heritability. If a model detects a significant QTL from such simulated data, this discovery is a false positive. As expected, functional mapping has small FPRs (<0.09) if the data were simulated under monoculture (Supplementary Table 3). For the co-culture data, small FPRs (<0.10) were also found by

CoCoM, even when a sample size is modest, suggesting that the new model has a reasonably high specificity.

## Discussion

In nature, a species can adapt not only to physical environments (e.g., temperature), but also to another species with which it interacts (e.g., competitors)[31]. It has well been recognized that interspecies interactions affect community composition and adaptation to changing environment[2,3] and operate as an evolutionary force that drives species to change through time and space[33–35]. Traditional genetic mapping focuses on the phenotypic variation of a single species, failing to characterize how QTLs determine multiple species as a community. In this article, we present a competition–cooperation mapping (CoCoM) model to unravel the genetic machineries of interspecies interactions.

The merit of CoCoM lies in its statistical synthesis of systems mapping and community ecology into a genetic setting. It can test and estimate how QTLs modulate interspecific competition and cooperation and interpret the critical roles of these QTLs in organizing community structure and function through mutualism, antagonism, parasitism, commensalism, and amensalism. In a co-culture experiment of *E. coli* and *S. aureus*, CoCoM detected a set of QTLs for microbial interactions, most of which determine an antagonistic relationship between these two bacterial species (Fig. 4). These QTLs expressed in a socialized environment were found to differ from those in a socially isolated environment (Fig. 2), suggesting that ecological interactions activate the expression of some unique genes. Taken together, CoCoM can glean new insight into the genetic architecture of interspecific interactions, species co-evolution, and community dynamics.

The genetic underpinning for the response of a species to abiotic factors only contains genes from this species, whereas the

biotic response of a species is also driven by genes of its coexisting species and the epistatic interaction between genes from different species[35,36]. As a reductionist approach, traditional QTL mapping can only identify direct genetic effects of QTLs from a species on its own phenotype, but cannot detect indirect genetic effects of QTLs from one species on the phenotype of its interacting partners in the same community and genome–genome epistatic genetic effects of QTLs from different species on the community phenotype. Epistasis may occur not only among the QTLs of the same genome, but also among the QTLs derived from different genomes. In molecular genetic experiments, such epistatic QTLs that act depending on the other genomes have been mapped[36] and the molecular pathways implicated for QTL actions identified[37]. CoCoM can separate direct, indirect, and interspecific genome–genome epistatic effects of QTLs involved in community composition and dynamics. In our co-culture experiment, indirect and genome–genome epistatic effects together contribute to more than one half of the total genetic variance for bacterial growth in a community.

CoCoM is a dynamic model that capitalizes on time series phenotypic data to search for interaction QTLs. To assess its advantage, we implemented and compared a static mapping model based on the phenotypic data measured at discrete time points. Previous statistical analyses have shown that, compared to a static model, a dynamic mapping model is biologically more relevant due to its embedment of biologically meaningful equations and statistically more powerful through parsimonious modeling of the mean covariance structures[13,14]. These advantages hold for the dynamic CoCoM model that detected much more biologically interpretable QTL pairs than its static counterpart (Fig. 2). Also, the dynamic model can visualize the temporal-spatial pattern of genetic effects and, therefore, should be particularly useful for inferring and predicting the genetic machineries of community dynamic and evolution.

How does interspecific epistasis play a role in the dynamic trajectories of ecosystem phenotypes[31]? The CoCoM model provides a platform for testing the genome–genome epistasis of eco-devo. In addition, in which manner do specific genes that influence species competition and co-evolution operate in natural communities? Through the implementation of a zero-isocline analysis of differential equations, CoCoM can quantitatively determine when one species should deplete all resources of its competing counterparts to survive, or it can coexist and, thus, coevolve with the others over ecological time in natural communities (Supplementary Fig. 2). It can further identify specific QTLs that govern this ecological process.

CoCoM represents a general framework for mapping community dynamics, for which there is much room to improve. First, the recent development of statistical variable selection has made it possible to visualize a network of genetic interactions among a large number of loci in genome-wide association studies[38]. The integration of CoCoM and variable selection will

enable the systematical characterization of direct, indirect, and genome–genome epistatic genetic effects throughout the entire genome. Second, a complex community is often composed of multiple species in which interspecies interactions are organized into a network[39]. A group of high-dimensional ordinary differential equations should be derived and implemented into CoCoM to quantify how each and every species interacts and communicates with all possible other species in a network and identify hub or keystone species that play a leadership role in mediating network dynamics. Third, additive Lotka–Volterra pairwise models can only characterize how one species stimulates or inhibits the abundance of other species in a gross way. By considering the chemical mediators underlying interspecific interactions, Momeni et al. theoretically showed that these additive models fail to capture the complexity of ecological interactions[40]. These authors have formulated mechanistic reference models for predicting two different species engaging in chemical-mediated interactions. By incorporating Momeni et al.'s mechanistic models, CoCoM can be armed to establish a more precise predictive model of community dynamics and evolution of interacting species in ecological systems.

## Methods

**Design of ecological experiment**. Microorganisms are thought to be ideal material for ecological experiments aimed to characterize interspecific interactions[41]. Suppose there are two microbial species A and B, from each of which we randomly sample $n$ strains to produce two natural mapping populations and genotype these samples for single-nucleotide polymorphisms (SNPs) throughout the entire genome for the two species. To study how two species compete in the same environment, we pair each strain from one species randomly with one (and only one) strain from the other, thus forming a total of $n$ independent pairs. Each pair is cultured in a separate flask with at least three replicates. The flasks are laid out randomly with no position effect in a laboratory.

Depending on how they can share the resource, two strains from different species within a flask may choose to compete or cooperate. We investigate such inner workings in each flask by measuring the abundance of each strain repeatedly at multiple times (say $T$) during microbial growth. We hypothesize that the abundance dynamics of each species is governed by QTLs from its own genome (through direct effects), QTLs from the genome of its coexisting species (through indirect effects), and interactions between the QTLs from different species (genome–genome epistasis).

**Quantitative genetic model**. Assume that there is a QTL on species A with two genotypes $A$ and $a$ and another QTL on species B with two genotypes $B$ and $b$. The two QTLs form four interspecific genotype combinations, $A/B$, $A/b$, $a/B$, and $a/b$, whose genotypic values for the abundance of each species $l$ ($l = $ A or B) at time $t$ are denoted as $\mu_{AB}^l(t)$, $\mu_{Ab}^l(t)$, $\mu_{aB}^l(t)$, and $\mu_{ab}^l(t)$, respectively. According to quantitative genetic theory, these genotypic values can be partitioned into their components[42], shown in Table 1. Let $\mu^A(t)$ and $\mu^B(t)$ denote the time-dependent population means of abundance for species A and B, respectively, $a_A^A(t)$ denotes the time-dependent direct genetic effect of species A's QTL on its own abundance, $a_B^B(t)$ denotes the time-dependent direct genetic effect of species B's QTL on its own abundance, $a_A^B(t)$ denotes the time-dependent indirect genetic effect of species A's QTL on the abundance of its coexisting species B, $a_B^A(t)$ denotes the time-dependent indirect genetic effect of species B's QTL on the abundance of its coexisting species A, and $I_{A \times B}^A(t)$ and $I_{A \times B}^B(t)$ denote the time-dependent genome–genome epistatic effect between the QTLs of two species on the abundance of species A and B, respectively.

**Table 1 Overall genotypic values of genome–genome combinations at a QTL and their decomposition into different components for two coexisting bacterial (haploid) species A and B**

| Genotype combination | | Genotypic value of abundance for different species | |
|---|---|---|---|
| **A** | **B** | **A** | **B** |
| $A$ | $B$ | $\mu_{AB}^A(t) = \mu^A(t) + a_A^A(t) + a_B^A(t) + I_{A \times B}^A(t)$ | $\mu_{AB}^B(t) = \mu^B(t) + a_A^B(t) + a_B^B(t) + I_{A \times B}^B(t)$ |
| $A$ | $b$ | $\mu_{Ab}^A(t) = \mu^A(t) + a_A^A(t) - a_B^A(t) - I_{A \times B}^A(t)$ | $\mu_{Ab}^B(t) = \mu^B(t) + a_A^B(t) - a_B^B(t) - I_{A \times B}^B(t)$ |
| $a$ | $B$ | $\mu_{aB}^A(t) = \mu^A(t) - a_A^A(t) + a_B^A(t) - I_{A \times B}^A(t)$ | $\mu_{aB}^B(t) = \mu^B(t) - a_A^B(t) + a_B^B(t) - I_{A \times B}^B(t)$ |
| $a$ | $b$ | $\mu_{ab}^A(t) = \mu^A(t) - a_A^A(t) - a_B^A(t) + I_{A \times B}^A(t)$ | $\mu_{ab}^B(t) = \mu^B(t) - a_A^B(t) - a_B^B(t) + I_{A \times B}^B(t)$ |

Subscripts denote genotype combinations between two species or the species from which the QTL is derived, and superscripts denote the species whose phenotype (abundance) is affected by the QTL

From expressions in Table 1, each of these genetic effect components are solved by

$$a_A^A(t) = \tfrac{1}{4}\big(\mu_{AB}^A(t) + \mu_{Ab}^A(t) - \mu_{aB}^A(t) - \mu_{ab}^A(t)\big) \tag{5a}$$

$$a_B^B(t) = \tfrac{1}{4}\big(\mu_{AB}^B(t) + \mu_{aB}^B(t) - \mu_{Ab}^B(t) - \mu_{ab}^B(t)\big) \tag{5b}$$

$$a_A^B(t) = \tfrac{1}{4}\big(\mu_{AB}^B(t) + \mu_{Ab}^B(t) - \mu_{aB}^B(t) - \mu_{ab}^B(t)\big) \tag{5c}$$

$$a_B^A(t) = \tfrac{1}{4}\big(\mu_{AB}^A(t) + \mu_{aB}^A(t) - \mu_{Ab}^A(t) - \mu_{ab}^A(t)\big) \tag{5d}$$

$$I_{A\times B}^A(t) = \tfrac{1}{4}\big(\mu_{AB}^A(t) + \mu_{ab}^A(t) - \mu_{Ab}^A(t) - \mu_{aB}^A(t)\big) \tag{5e}$$

$$I_{A\times B}^B(t) = \tfrac{1}{4}\big(\mu_{AB}^B(t) + \mu_{ab}^B(t) - \mu_{Ab}^B(t) - \mu_{aB}^B(t)\big) \tag{5f}$$

All these direct, indirect, and genome–genome epistatic effects jointly construct the genotype–phenotype map, although previous quantitative genetic studies can only estimate the direct effects.

**Mixture model and estimation**. We formulate static and dynamic models to estimate QTL genotype-specific parameters by associating marker data and phenotypic data for $n$ interspecific pairs of strains. Let $N_i^A(t)$ and $N_i^B(t)$ denote the abundance of individuals from species A and B in a pair ($i$) of strains at time $t$ ($t = 1, ..., T$). A mixture-based likelihood model has been widely used to map QTLs for complex traits[42]. Considering microbial abundance data at time $t$, the likelihood is written as

$$L(\mathbf{N}^A(t), \mathbf{N}^B(t)) = \prod_{i=1}^n \Big[ \omega_{A|Bi} f_{AB}(N_i^A(t), N_i^B(t)) + \omega_{A|bi} f_{Ab}(N_i^A(t), N_i^B(t)) \\ + \omega_{a|Bi} f_{aB}(N_i^A(t), N_i^B(t)) + \omega_{a|bi} f_{ab}(N_i^A(t), N_i^B(t)) \Big] \tag{6}$$

where $\omega_{\bullet|i}$ is the conditional probability of a particular interspecific QTL genotype combination given the marker genotype combination of interspecific pair $i$, which is expressed by marker-QTL linkage disequilibria in the populations of two species;[42] and $f_\bullet(N_i^A(t), N_i^B(t))$ is a bivariate normal distribution function of species A and B with genotype combination-dependent mean vector $\big(\mu_\bullet^A(t), \mu_\bullet^B(t)\big)$ and covariance matrix composed of species A's variance $(\sigma_A^2(t))$, species B's variance $(\sigma_B^2(t))$ and species–species correlation $(\rho(t))$. Based on the maximum likelihood estimates of $\big(\mu_\bullet^A(t), \mu_\bullet^B(t)\big)$, we can estimate the direct, indirect, and genome–genome epistatic effects according to Eqs. (5a)–(5f).

The mixture-based likelihood of time-dependent abundance data for $n$ interspecific pairs is expressed as

$$L(\mathbf{N}^A, \mathbf{N}^B) = \prod_{i=1}^n \Big[ \omega_{AB|i} f_{AB}(\mathbf{N}_i^A, \mathbf{N}_i^B; \mathbf{\Theta}_{AB}, \mathbf{\Psi}) + \omega_{Ab|i} f_{Ab}(\mathbf{N}_i^A, \mathbf{N}_i^B; \mathbf{\Theta}_{Ab}, \mathbf{\Psi}) \\ + \omega_{aB|i} f_{aB}(\mathbf{N}_i^A, \mathbf{N}_i^B; \mathbf{\Theta}_{aB}, \mathbf{\Psi}) + \omega_{ab|i} f_{ab}(\mathbf{N}_i^A, \mathbf{N}_i^B; \mathbf{\Theta}_{ab}, \mathbf{\Psi}) \Big] \tag{7}$$

where $\mathbf{N}_i^A = \big(N_i^A(1), ..., N_i^A(T)\big)$ and $\mathbf{N}_i^B = \big(N_i^B(1), ..., N_i^B(T)\big)$ are the vectors of abundance trajectories at $T$ times for species A and B, respectively, and $f_\bullet(\mathbf{N}_i^A, \mathbf{N}_i^B; \mathbf{\Theta}_\bullet, \mathbf{\Psi})$ is a multivariate normal distribution with expected mean vector for pair $i$ that belongs to a particular interspecific QTL genotype combinations, expressed as

$$\mu_{AB} = (\mu_{AB}^A; \mu_{AB}^B) \equiv (\mu_{AB}^A(1), ..., \mu_{AB}^A(T); \mu_{AB}^B(1), ..., \mu_{AB}^B(T)) \tag{8a}$$

$$\mu_{Ab} = (\mu_{Ab}^A; \mu_{Ab}^B) \equiv (\mu_{Ab}^A(1), ..., \mu_{Ab}^A(T); \mu_{Ab}^B(1), ..., \mu_{Ab}^B(T)) \tag{8b}$$

$$\mu_{aB} = (\mu_{aB}^A; \mu_{aB}^B) \equiv (\mu_{aB}^A(1), ..., \mu_{aB}^A(T); \mu_{aB}^B(1), ..., \mu_{aB}^B(T)) \tag{8c}$$

$$\mu_{ab} = (\mu_{ab}^A; \mu_{ab}^B) \equiv (\mu_{ab}^A(1), ..., \mu_{ab}^A(T); \mu_{ab}^B(1), ..., \mu_{ab}^B(T)) \tag{8d}$$

and covariance matrix

$$\mathbf{\Sigma} = \begin{pmatrix} \mathbf{\Sigma}_A & \mathbf{\Sigma}_{AB} \\ \mathbf{\Sigma}_{BA} & \mathbf{\Sigma}_B \end{pmatrix} \tag{9}$$

with $\mathbf{\Sigma}_A$ and $\mathbf{\Sigma}_B$ being ($T \times T$) covariance matrices of abundances over time and $\mathbf{\Sigma}_{AB} = \mathbf{\Sigma}_{BA}^T$ being a ($T \times T$) covariance matrix between two species.

Systems mapping models genotypic value vectors of each interspecific QTL genotype combination, i.e., $\mu_{AB}$, $\mu_{Ab}$, $\mu_{aB}$, and $\mu_{ab}$, by a group of LV-based ODEs (1) characterized by interspecific QTL genotype combination-dependent parameters $\mathbf{\Theta}_{AB}$, $\mathbf{\Theta}_{Ab}$, $\mathbf{\Theta}_{aB}$, and $\mathbf{\Theta}_{ab}$, respectively. For all genotype combinations, systems mapping assumes the same covariance matrix $\mathbf{\Sigma}$, modeled by a set of

matrix-structuring parameters $\mathbf{\Psi}$. As shown in Fu et al.[43], the fourth-order Runge–Kutta algorithm can be implemented to solve the differential equations within the mixture model framework, which obtains the maximum likelihood estimates of $\mathbf{\Theta}_{AB}$, $\mathbf{\Theta}_{Ab}$, $\mathbf{\Theta}_{aB}$, and $\mathbf{\Theta}_{ab}$. The covariance structure is modeled by using a parsimonious and flexible approach, such as autoregressive, antedependence, autoregressive moving average, or nonparametric and semiparametric approaches. These approaches have been used and tested in the functional mapping and systems mapping[44]. The EM algorithm is implemented to estimate marker-QTL haplotype frequencies[42], hybridized with the Runge–Kutta algorithm to estimate $\mathbf{\Theta}_{AB}$, $\mathbf{\Theta}_{Ab}$, $\mathbf{\Theta}_{aB}$, and $\mathbf{\Theta}_{ab}$ and the simplex algorithm to estimate covariance-structuring parameters $\mathbf{\Psi}$.

**Hypothesis testing**. Based on static likelihood (6) and dynamic likelihood (7), we can test whether there are significant interspecific QTLs involved in interspecific interactions. Using the dynamic model as an example, this can be done by formulating the two hypotheses:

$$\begin{aligned} &H_0: \Theta_{AB} = \Theta_{Ab} = \Theta_{aB} = \Theta_{ab} \equiv \Theta \\ &H_1: \text{Not all equalities in the } H_0 \text{ hold} \end{aligned} \tag{10}$$

under each of which the likelihoods are calculated, respectively. Then, their log-likelihood ratio is calculated and compared against a genome-wide critical threshold determined from permutation tests or score statistics[45]. If the null hypothesis above is rejected, this means that QTLs from two species have been detected by the molecular marker.

After significant QTLs are detected, the next is to test whether these QTLs exert significant direct effects, indirect effects, and interspecific genome–genome epistatic effects. The null hypotheses for these tests are, respectively, expressed as

$$H_0: a_A^A(t) = 0 \text{ and } a_B^B(t) = 0 \text{ for direct effects} \tag{11}$$

$$H_0: a_A^B(t) = 0 \text{ and } a_B^A(t) = 0 \text{ for indirect effects} \tag{12}$$

$$H_0: I_{A\times B}^A(t) = 0 \text{ and } I_{A\times B}^B(t) = 0 \text{ for genome} - \text{genome epistatic effects} \tag{13}$$

The critical thresholds for tests (11)–(13) can be obtained from simulation studies. Two effects of each test on the abundance of species A and B, respectively, can be further tested for their significance.

**Study material**. The model was validated by an experiment of bacterial competition. We collected strains of *E. coli* and *S. aureus*, whose IDs were listed in Supplementary Table 4, from National Infrastructure of Microbial Resources, China, and paired these strains between species to form 45 independent interspecific combinations. Each pair was co-cultured with a 1:1 ratio in a separate flask but with the same media for all pairs. Meanwhile, all paired strains were mono-cultured individually under the same condition. Cultures were established in 50 mL flasks containing 25 mL of two-times diluted brain heart infusion medium (OXOID, Basingstoke, England) and inoculated initially from established cultures of bacteria after 4-day cultivation. In the co-culture treatment, inoculates of each species were added to the same flask to create a two-species community. The starting concentration of each species was $5 \times 10^3$ copies/mL. In the monoculture treatment, bacteria were cultured in 25 mL medium with a starting concentration of $5 \times 10^3$ copies/mL. We replicated the experiment three times for both mono-culture and co-culture.

We used quantitative PCR (qPCR) to count the number of cells for each strain in each flask at time once every 2 h before 24 h of culture, followed by once every 4 h after 24 h. An Mx3005P real-time system (Stratagene, La Jolla, USA) was used to perform qPCR in a total volume of 25 µL, consisting of the SuperReal PreMix Plus (SYBR Green) (TIANGEN, Beijing, China), 300 nM forward primers and 300 nM reverse primers. Genomic DNA was extracted by the TIANamp Bacteria DNA Kit. For specific detection of *E.coli* species, 217 bp of the regulatory region of *uidA* gene, designated *uidR*, which is located upstream of the *uidA* structural gene, were amplified by forward primer GTGGCAGTGAAGGGCGAACAGT and reverse primer GTGAGCGTCGCAGAACATTACA. For specific detection of *S. aureus* species, 226 bp of *nuc* gene encoding thermostable nuclease, were amplified by forward primer AAAGGGCAATACGCAAAGAGGT and reverse primer CTTTAGCCAAGCCTTGACGAAC. Control samples, without template DNA, were also included in the runs. The thermal cycling conditions were as follows: an initial denaturation at 95 °C for 10 min followed by 40 cycles of 30 s at 95 °C, 1 min at 55 °C, and 1 min at 72 °C. Each run ended with a melting curve analysis. Fluorescence data were collected at the end of each cycle and determination of the cycle threshold line was carried out automatically by the instrument. The DNA copy number of each species was calculated using a *uidA*/*nuc*-containing plasmid of known concentration as a standard. The qPCR counts of each strain/pair, averaged over three replicates, were used for genetic mapping.

Whole-genome sequencing was performed on the Illumina HiSeq2000 platform at Novogene (Novogene, Beijing, China) using E. coli str. K-12 substr.

MG1655 and S. aureus subsp. aureus NCTC 8325 as the reference strain, respectively. Average sequencing depth and coverage for E. coli and S. aureus were summarized in Supplementary Table 4. Illumina reads were mapped directly to the E. coli and S. aureus reference sequences using BWA mapper (Version 0.7.8). In alignment results, PCR duplicates were removed by SAMtools software package (Version 0.1.18). We also used SAMtools to detect SNPs. Every time a mapped read shows a mismatch from the reference genome, SAMtools can be used to figure out whether the mismatch is due to a real SNP. It incorporates different types of information, such as the number of different reads that share a mismatch from the reference, sequence quality data, and expected sequencing error rates, which helps to determine how observed mismatches occur. SNPs with high-quality score ($Q$ value ≥20) and enough supporting bases ≥4) (with the variant) were kept as final SNPs result. In total, 168,720 SNPs that cover the E. coli genome by approximately one per 23 bp and 83,642 SNPs that cover the S. aureus genome by one per 41 bp were obtained for these strains. These SNP densities should be sufficient enough for the identification of genomic regions in a genome-wide association study.

The study material was subject to structural analysis using Q-ROADTRIPS[46,47]. It identified five and eight subpopulations among the E. coli and S. aureus strains sampled, respectively. The subsequent association studies were based on the phenotypes that have been adjusted for these subpopulation differences. As shown by Q–Q plots (Supplementary Fig. 7), the confounding effects of population structure have been well removed for functional mapping of each species growth in monoculture and systems mapping of two species in co-culture.

**Code availability**. The computer code is available as an open R package at https://github.com/LiboJiang/MicrobialInteraction or under request directly from the corresponding author.

**Data availability**. The raw sequence data were deposited in the NCBI short reads archive under accession number SRP074089 and SRP074912. The other data are available with the computer code at https://github.com/LiboJiang/MicrobialInteraction, or under request directly from the corresponding author.

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

## Acknowledgements

This work is supported by Fundamental Research Funds for the Central Universities (No. 2015ZCQ-SW-06, 2017JC05), National Natural Science Foundation of China

(grant 31700576), the "one-thousand person" plan, grant U01 HL119178, and grant NICHD 5R01HD086911-02.

## Author contributions

R.W. conceived the conceptual idea, supervised the study, and wrote the manuscript; L.J. derived the model, performed data analysis and simulation studies, and wrote R-based software. X.H. and Y.J. designed the experiment and wrote the study material part of the manuscript. M.Y. and M.S. participated in computer simulation and data analysis. N.C., J.Z., Z.Z., and J.L. participated in data collection.

## Additional information

**Competing interests:** The authors declare no competing interests.

