## [Peer Review File · Nature Communications]

Reviewers' comments:

Reviewer #1 (Remarks to the Author):

The paper by Jiang et al reports a new method to infer inter-species selective effects from time-series data of microbial population dynamics in a multi-strain community. Such data are fitted to a Lotka-Volterra (LV) model, and cross-species fitness interactions, including epistasis are inferred by a probabilistic method. The paper is potentially very interesting as method development for inference of species interactions in complex communities, such as microbiota. However, there are central points of the method which I do not fully understand, preventing me from assessing its consistency, as well as concerns of data overfitting. I think these concerns should be addressed in a revision before the paper qualifies for Nature Communications. I have the following specific questions:

1. General method description:

(a) A number of terms are not clearly defined. What is the relation of the μ variables to the abundances N in the main text? What exactly is "overall mean" (line 427)? What are genetic effects (fitness?), how are they related to the LV couplings α in the main text?

2. Most importantly, why are the effects a in eqs. (2), (3), (4) taken as time-dependent? In a standard LV approach, the α coefficients would be constant or more slowly varying than the population numbers N . Maybe it would help to carefully distinguish between the fitness interactions between strains and the contribution of individual genetic loci to these interactions? I understand that the latter can show a stronger time dependence, but do we learn anything about the former? This could be tested by simulations with known time-independent, allele-specific species couplings.

3. The probabilistic inference, eq. (5) and below, should be described in more detail. What is the error model underlying the likelihood function (5)? If it is measurement error, what are the underlying assumptions on observations, why is genetic drift neglected?

4. Why is (1) taken as a separate dynamical equation? Should it not be contained in the LV equations (2)?

5. Another important concern is the apparently independent inference of the QTL effects of different genomic loci. The microbial populations are presumably under clonal interference, which implies strong correlations of LD between different loci; see, e.g. Illingworth et al, Quantifying selection in evolving populations using time-resolved genetic data, J. Stat. Mech. 2013.

6. Some of the language in the introduction contains a lot of grand jargon I suggest to remove. For example, "Thus, the holistic behavior of the system, i.e., the phenotype of the trait studied, can be quantified and predicted". I am not sure what holistic is supposed to mean, phenotype and trait are often synonymous, and the model provides an inference of couplings but no predictions. The Lotka-Volterra model is very well known, including its form as coupled ODEs, so it would suffice to emphasize that the paper wants to infer LV species couplings and their genetic basis from time-series of abundance. Which would be no small matter if successful.

Reviewer #2 (Remarks to the Author):

Jiang, He, Jin et al. provide a genetic model and estimation procedure for mapping QTL that affect genetically complex traits when multiple species are present in a mixed system, and apply their model to growth data from two bacterial species. This contribution will be of interest to the field, and it may find additional use in areas beyond ecological dynamics. The paper is clearly structured and written. There are some minor problems with grammar and word choice, but I was able to

follow the content relatively easily.

Specific comments:

Major comments:

1. Computer code implementing this method should be made available. I did not find information about code availability in the manuscript.
2. The methods on the experimental growth data are not sufficient. Please provide detail on these items:
 - a) Did you perform replicates for culture growth? How reproducible are the measurements for a given pair? With so much variation among strain pairs, and with so much variation in how each strain responds to co- vs. monoculture, a natural question is how much of this variation is just random noise.
 - b) The information on the sequencing and bioinformatics pipeline used to call SNPs in the 45 strains is insufficient. E.g. how deep was the sequencing, which alignment software was used, and how were SNPs filtered? Please provide additional detail on this pipeline.
 - c) It is not clear if the growth data was obtained in liquid culture or on solid media.
 - d) There is no information on which media was used to grow the strains.
 - e) describe how the qPCR was performed. Which genomic target? Which primer sequences? Validated how? Which reagents and cycling conditions? How were qPCR analyses conducted?
 - f) How were the monocultures measured – presumably not by qPCR?
 - g) How was DNA extracted for qPCR?

Minor comments:

3. l. 103: "GO analysis has annotated a larger portion of significant loci detected to specific genome positions of candidate genes, [...]"
I do not understand this sentence – please reword.
4. l. 107 "A tremendous difference in QTL detection between monoculture and co-culture". I'm not sure what the authors mean by this "tremendous" difference. The numbers of QTL were not that different between mono- and coculture (especially in *S. aureus*). Do the authors mean to say that the QTL that are found for mono vs. co-culture are in different locations? Please clarify.
5. In the strategy matrix (3), are commensalism and amensalism flipped? E.g. the middle entry in the leftmost column is "commensalism", although one species has no effect on the 2nd, while the 2nd species is hurt (negative $N_{s<-e}$) by the 1st. Please carefully check the table and the definitions below it.
6. In figure 3A, what are the many grey dots?

Reviewer #1

The paper by Jiang et al reports a new method to infer inter-species selective effects from time-series data of microbial population dynamics in a multi-strain community. Such data are fitted to a Lotka-Volterra (LV) model, and cross-species fitness interactions, including epistasis are inferred by a probabilistic method. The paper is potentially very interesting as method development for inference of species interactions in complex communities, such as microbiota. However, there are central points of the method which I do not fully understand, preventing me from assessing its consistency, as well as concerns of data overfitting. I think these concerns should be addressed in a revision before the paper qualifies for Nature Communications. I have the following specific questions:

Our response: Thank you for your positive and constructive comments on the utility of our new model. The Lotka-Volterra (LV) model has proven very powerful to describe the interactive relationships between different organisms growing or grown in the shared environment. It is natural to use the LV to fit the dynamic change of microbial growth in our experiment of co-culture. However, as pointed out by this reviewer, data overfitting, if existing, would lead to biased results. To examine if our fitting using the LV model is overfitting, we have added a new figure (current Fig. 3) in which overfitting was shown not to occur.

1. General method description:

(a) A number of terms are not clearly defined. What is the relation of the μ variables to the abundances N in the main text? What exactly is "overall mean" (line 427)? What are genetic effects (fitness?), how are they related to the LV couplings α in the main text?

Our response: We understand that a clear description of terms is the key for a methods paper. For this reason, we appreciate the reviewer's comment and have paid special attention to clarifying the terms used.

- The relation of μ variables and abundance N : N is the abundance of strains which is just a phenotypic trait, and μ is the genotypic value of abundance for a QTL genotype. For example, there are 10 strains with abundance N_1, N_2, \dots, N_{10} , the first five of which are genotype AA and the last five of which are genotype aa. The genotypic value of AA is $\mu_{AA} = (N_1 + \dots + N_5)/5$, and the genotypic value of aa is $\mu_{aa} = (N_6 + \dots + N_{10})/5$.
- Overall mean: It is also called the population mean in quantitative genetics. In this version, we call it the population mean.
- Genetic effect: It is expressed as the effect triggered by a QTL. Using the above example, the genetic effect of this QTL is calculated as $a = \mu_{AA} - \mu_{aa}$ (the difference of genotypic values between two genotypes at a QTL).
- Link between genetic effects and LV equations. The innovation of this paper just lies in the elegant link between these two things. When strains from one species are paired with strains from the second species, we have multiple interspecific pairs. Consider a QTL with genotypes AA and aa for one species and a QTL with genotypes BB and bb for the second species. Thus, all these interspecific pairs will have four combinations of genotypes from different species, i.e., AA/BB, AA/bb, aa/AA, and aa/bb. We used the LV equation to fit the abundance of strains from each of these combinations for two species, obtaining the estimates of α and

the mean curves of abundance for each combination. By using the equations (5a) – (7b), we can estimate various genetic effect curves. I hope this explanation has addressed your question.

2. Most importantly, why are the effects α in eqs. (2), (3), (4) taken as time-dependent? In a standard LV approach, the α coefficients would be constant or more slowly varying than the population numbers N . Maybe it would help to carefully distinguish carefully between the fitness interactions between strains and the contribution of individual genetic loci to these interactions? I understand that the latter can show a stronger time dependence, but do we learn anything about the former? This could be tested by simulations with known time-independent, allele-specific species couplings.

Our response: We address these comments piecemeal.

- “... why are the effects α in eqs. (2), (3), (4) taken as time-dependent?”: The effects are re-numbered as eqn (5a) – (7b) in the revision. As explained above, genotypic curves can be estimated by fitting the LV model to raw data for each genotypic combination. Because these curves are time-dependent, we can obtain the time-dependent estimates of genetic effects by using eqn (5a) – (7b).
- “In a standard LV approach, the α coefficients would be constant or more slowly varying than the population numbers N ”: This is true because the current form of the LV model used is deterministic. In the stochastic version, α coefficients can be time-varying. This would be another story to explore.
- “Maybe it would help to carefully distinguish carefully between the fitness interactions between strains and the contribution of individual genetic loci to these interactions?”: Yes, this is just a central theme of this paper. Different strains may interact with one another when they are co-cultured. QTLs may affect the strength and direction of these interactions. We aim to find these QTLs by integrating systems mapping and game theory.
- “I understand that the latter can show a stronger time dependence, but do we learn anything about the former?”: In fact, the former, i.e., the fitness interactions between species, has been very well explored in the ecological and evolutionary literature. However, little is known about the latter. Our aim is just to identify QTLs for such interactions.
- “This could be tested by simulations with known time-independent, allele-specific species couplings“: This is a good point. We have performed simulation studies to resolve this issue (see the text). By simulating the data with known time-independent patterns for each genotype combination under different residual errors, we have tested and validated the statistical power of how our model can identify interaction QTLs.

3. The probabilistic inference, eq. (5) and below, should be described in more detail. What is the error model underlying the likelihood function (5)? If it is measurement error, what are the underlying assumptions on observations, why is genetic drift neglected?

Our response: This equation, now renumbered as eqn (8), is a mixture-based likelihood model. This likelihood contains four mixture components, each representing an interspecific QTL genotype combination, A/B , A/b , a/B , and a/b , weighted by their relative probabilities. This model is regarded as a standard model for QTL mapping. A reference (45) was cited to reflect its popularity.

The probability density function for each mixture component in eqn (8) is specified by a multivariate normal distribution with the mean vectors (depending on which QTL genotype combination is considered) and covariance matrix. The phenotypic value y_i for pair i is written as

$$y_i = x_i \mu_j + e_i$$

where x_i is the genotypic value of a particular QTL genotype combination j , x_i is the indicator variable that is expressed as 1 if the pair belongs to combination j and 0 otherwise, and e_i is the residual error of pair i . This residual error include measurement error, experimental noise, genetic drift and etc, which is assumed to follow a multivariate normal distribution. With this assumption, we constructed a residual covariance matrix (10) that reflects the longitudinal structure of time-dependent variances and covariances. We showed that the covariance structure is modeled by using a parsimonious and flexible approach, such as autoregressive, antedependence, autoregressive moving average, or nonparametric and semiparametric approaches. These approaches have been used and tested in the functional mapping and systems mapping (ref. 47).

4. Why is (1) taken as a separate dynamical equation? Should it not be contained in the LV equations (2)?

Our response: Eqn (1) is a univariate growth equation that describes logistic growth for a strain in monoculture, whereas the LV equation reflects the interactive growth of two strains co-cultured in the same environment. These two equations were used to compare the difference of growth trajectories for strains in monoculture (with no competition) and co-culture (with competition). In fact, the LV equation contains the logistic growth component. As such, they have some similarity: the LV can be reduced to eqn (1) when strains are cultured in isolation.

5. Another important concern is the apparently independent inference of the QTL effects of different genomic loci. The microbial populations are presumably under clonal interference, which implies strong correlations of LD between different loci; see, e.g. Illingworth et al, Quantifying selection in evolving populations using time-resolved genetic data, J. Stat. Mech. 2013.

Our response: This is a good point. The purpose of this paper is to present a new model for QTL mapping using interactive data. For this reason, we attempt to identify QTLs one by one. Presently, variable selection has been implemented to analyze all markers at the same time (ref 41), which can powerfully handle all possible correlations among different loci through LD. This issue has been discussed in the Discussion.

6. Some of the language in the introduction contains a lot of grand jargon I suggest to remove. For example, "Thus, the holistic behavior of the system, i.e., the phenotype of the trait studied, can be quantified and predicted". I am not sure what holistic is supposed to mean, phenotype and trait are often synonymous, and the model provides an inference of couplings but no predictions. The Lotka-Volterra model is very well known, including its form as coupled ODEs, so it would suffice to emphasize that the paper wants to infer LV species couplings and their genetic basis from time-series of abundance. Which would be no small matter if successful.

Our response: We have very carefully clarified jargons. Sentence "Thus, the holistic behavior of the system, i.e., the phenotype of the trait studied, can be quantified and predicted" has been modified as "Thus, the phenotype, here interpreted as the behavior of the system, can be quantified by the ODEs."

- "Holistic" was removed.
- "Trait" was removed, because "trait" are synonymous to "phenotype"
- "Predicted" was removed. We agree with this Reviewer that the LV model provides an inference of species couplings rather than its prediction. This model has shown to be powerful for fitting the abundance of species in our study and other studies.

Reviewer #2

Jiang, He, Jin et al. provide a genetic model and estimation procedure for mapping QTL that affect genetically complex traits when multiple species are present in a mixed system, and apply their model to growth data from two bacterial species. This contribution will be of interest to the field, and it may find additional use in areas beyond ecological dynamics. The paper is clearly structured and written. There are some minor problems with grammar and word choice, but I was able to follow the content relatively easily.

Our response: Thank you for your encouragement to our work.

Specific comments:

Major comments:

1. Computer code implementing this method should be made available. I did not find information about code availability in the manuscript.

Our response: The data and computer code that support the findings of this study are available from the authors upon request or can be downloaded freely from <http://ccb.bjfu.edu.cn/program.html>.

2. The methods on the experimental growth data are not sufficient. Please provide detail on these items:

a) Did you perform replicates for culture growth? How reproducible are the measurements for a given pair? With so much variation among strain pairs, and with so much variation in how each strain responds to co- vs. monoculture, a natural question is how much of this variation is just random noise.

Our response: All these questions are reasonable. Thank this reviewer for asking.

- "Did you perform replicates for culture growth? How reproducible are the measurements for a given pair?": Yes, we performed three replicates for both monoculture and co-culture. The qPCR results of each strain/pair were the means over three independent experiments. We provide raw data for each replicate at <http://ccb.bjfu.edu.cn/program.html>. All this has been made clear in this revision.

- " With so much variation among strain pairs, and with so much variation in how each strain responds to co- vs. monoculture, a natural question is how much of this variation is just random noise": To test whether residuals are from random noise or random measurement errors, we plotted the estimated residuals (y) against the predicted values of abundance (x) for each strain pair by the LV model (please see Fig. 3 of the revision). It turns out that x and y has no correlation, confirming the randomness of residual errors.

b) The information on the sequencing and bioinformatics pipeline used to call SNPs in the 45 strains is insufficient. E.g. how deep was the sequencing, which alignment software was used, and how were SNPs filtered? Please provide additional detail on this pipeline.

Our response: We have added more details about the whole genome sequencing and SNP calling into the Study material and pre-preparation section.

c) It is not clear if the growth data was obtained in liquid culture or on solid media.

Our response: The growth data was obtained in liquid culture. Please the Study material and pre-preparation section.

d) There is no information on which media was used to grow the strains.

Our response: Brain Heart Infusion Medium (OXOID, Basingstoke, England) was chose to grow the strains. Please the Study material and pre-preparation section for the detail.

e) describe how the qPCR was performed. Which genomic target? Which primer sequences? Validated how? Which reagents and cycling conditions? How were qPCR analyses conducted?

Our response: We have added more details about how the qPCR was performed into Please the Study material and pre-preparation section.

f) How were the monocultures measured—presumably not by qPCR?

Our response: To keep the consistency of the data, the monocultures were also measured by the qPCR.

g) How was DNA extracted for qPCR?

Our response: Genomic DNA was isolated using the TIANamp Bacteria DNA Kit (TIANGEN, Beijing, China) according to the manufacturer's instructions. We have added the details of DNA extraction into Please the Study material and pre-preparation section.

Minor comments:

3. 1. 103: "GO analysis has annotated a larger portion of significant loci detected to specific genome positions of candidate genes, [...]"

I do not understand this sentence – please reword.

Our response: We have revised this sentence as " Results from GO analysis show that a large portion of QTLs detected by functional mapping reside in genome positions of candidate genes, ...". Please note that this sentence has been moved to the middle of this paragraph, which is more logic.

4. 1. 107 “A tremendous difference in QTL detection between monoculture and co-culture”. I’m not sure what the authors mean by this “tremendous” difference. The numbers of QTL were not that different between mono- and coculture (especially in *S. aureus*). Do the authors mean to say that the QTL that are found for mono vs. co-culture are in different locations? Please clarify.

Our response: Thanks for asking this question, which has been made clear in the revision. Yes, differences include those in QTL number and QTL location.

5. In the strategy matrix (3), are commensalism and amensalism flipped? E.g. the middle entry in the leftmost column is “commensalism”, although one species has no effect on the 2nd, while the 2nd species is hurt (negative $N_{s<-e}$) by the 1st. Please carefully check the table and the definitions below it.

Our response: Thank you for pointing this typo. We have corrected it.

6. In figure 3A, what are the many grey dots?

Our response: Each grey dot presents the log-likelihood ratio test statistic calculated to test whether a combination of QTLs from two species is statistically significant. The red dots denote those that are significant.

We have made some other editorial changes as needed. All changes were highlighted in red.

Reviewers' comments:

Reviewer #1 (Remarks to the Author):

The revision has addressed most of my comments in a satisfactory way, and I can now recommend the paper for publication. In particular, the simulations and the additional text on the statistical methods have improved the paper.

I would recommend, however, that the authors go one more time over the main text with the aim of improving clarity and conciseness. Three concrete examples: (a) I suggest to remove the text ``to downstream the phenotype. Thus, the phenotype, here interpreted as the behavior of the system, can be quantified by the ODEs. `` Downstream and behaviour are ambivalent terms in this context. (b) The ODE approach and its link to evolutionary game theory are well known (see, e.g. the book by Hofbauer and Sigmund). The label SEGM could be directed more clearly to the actual novelty of the paper, which is to learn map such interactions to genetic loci. (c) The term development might better be avoided, I am not sure what the ecological focus of the paper has to do with development.

Reviewer #2 (Remarks to the Author):

All my comments were addressed.

Reviewer #3 (Remarks to the Author):

This manuscript seeks to identify bacterial loci related to growth patterns in monoculture and co-culture. The authors have developed this new method, conducted some simulation studies using it, and used it with empirical growth data along with novel draft genome sequencing of 45 isolates each of the two focal strains.

I have a few fundamental, major concerns with the manuscript that preclude my ability to recommend it for publication, along with many minor comments detailed as highlighted passages and comments in the attached Word file.

- 1) I am not convinced that the results are not simply the result of over-fitting, given that the authors make everything in the model time-dependent. It would be more rigorous to first fit a non-time-dependent model (fixed parameters, with direct and epistatic genetic effects on them) before introducing the enormous degrees of freedom granted by time-dependence.
- 2) The relationship of the new method to previous methods is not well-explained. In a few places, comparisons were made between the new method and a previous method, but other than the new method having much smaller p-values (of which I am skeptical, due to comment 1), it is not clear what objective criteria were employed to compare the approaches.
- 3) The simulation portion of the study similarly lacked a rigorous presentation--how many loci contributed to the phenotype? What was the population structure of the simulated genomes? Were type I and type II error calculated and if so what were they as a function of sample size?
- 4) It was not clear whether appropriate multiple-testing corrections were employed. These are relatively small numbers of genomes, but for loci of strong effect it is possible to obtain significant genotype-phenotype correlations after Bonferroni correction or FDR. Please clarify this important point--using a nominal p-value of 5% would be predicted to give more hits than observed, given that there are ~100K SNPs per genome.

5) The conclusion that the genes identified were biologically meaningful based on the fact that they had annotation did not make sense to me. Was a formal GO-enrichment or other functional analysis conducted?

6) The relationship of the method and "game theory" was unclear. There did not seem to be any payoff matrices or fitness functions as in classical game theory or its modern descendent, adaptive dynamics. Furthermore, these short-term growth experiments would not have much opportunity for actual fitness, not was there competition between alternative alleles or phenotypes. Please clarify this connection or leave it out as it seems to promise something that is not delivered.

7) I believe there is a logical error in assuming that all pairs of species in fact are able to coexist with one another. Since this is never explicitly tested, all of the discussion about coexistence ought to be severely qualified or removed.

8) In batch culture, Michaelis-Menton or Gompertz growth functions can sometimes fit better than logistic growth. These alternative forms should be explored, particularly since there do not seem to be any conceptual reasons for preferring the logistic form with interaction coefficients. In particular, if cross-feeding is occurring this could be modeled explicitly. However, if interference competition is present then perhaps the LV form would in fact be the best model. In any case, this should be assessed and justified.

9) Overall I felt that the manuscript could be better structured so that it was clear what exactly the authors did and what precise outcomes were obtained, rather than having everything in the methods it would help to have some framing in the main text. The claims also seemed to be overstated and not clearly justified by the results.

Even with these concerns, I do think that applying GWAS methods (which appears to be what they actually did, rather than QTL mapping?) to ecological data such as these is very interesting, though it does not seem to have any connection to game theory. I would strongly suggest taking a step back and starting with time-independent parameters as the phenotypes rather than the abundances at each time point.

Very best wishes,
Maren L. Friesen

Reviewer #1

The revision has addressed most of my comments in a satisfactory way, and I can now recommend the paper for publication. In particular, the simulations and the additional text on the statistical methods have improved the paper.

I would recommend, however, that the authors go one more time over the main text with the aim of improving clarity and conciseness.

Our response: Yes, we have read over and think over again and again the text and have done our best to make our description clear and concise as much as possible.

Three concrete examples: (a) I suggest to remove the text “to downstream the phenotype. Thus, the phenotype, here interpreted as the behavior of the system, can be quantified by the ODEs. “ Downstream and behaviour are ambivalent terms in this context.

Our response: We have adopted this suggestion.

(b) The ODE approach and its link to evolutionary game theory are well known (see, e.g. the book by Hofbauer and Sigmund). The label SEGM could be directed more clearly to the actual novelty of the paper, which is to learn map such interactions to genetic loci.

Our response: This is a good point. Combining the third reviewer’ comment (please see below), we have considered this point by focusing on quantifying and visualizing the genetic architecture of interspecies interactions as the novelty of the paper.

(c) The term development might better be avoided, I am not sure what the ecological focus of the paper has to do with development.

Our response: Through deliberating this issue, we feel this reviewer is right. For this particular paper, “development” is not the most appropriate term because we focus on interspecific interactions as a force of community dynamic and evolution. Thus, we have changed “development” for “dynamics”.

Reviewer #3

This manuscript seeks to identify bacterial loci related to growth patterns in monoculture and co-culture. The authors have developed this new method, conducted some simulation studies using it, and used it with empirical growth data along with novel draft genome sequencing of 45 isolates each of the two focal strains.

Our response: First of all, we all are highly grateful for Dr. Friesen for her tremendous time and effort given to assess our manuscript. Her comments are very thoughtful and thorough. We have extremely carefully considered each of her comments and incorporated them into the manuscript. During this revising process, we have greatly benefited from Dr.

Friesen's comments. As can be seen, the manuscript has become much stronger and more readable after the incorporation and adoption of these comments.

The motivation of this manuscript is to develop a new conceptual framework for mapping genetic loci that mediate ecological interactions in a community. The major contributions of this work lie in two aspects: (1) Ecological interactions among different species are a driving force of community dynamics and evolution. The framework developed is among the first statistical models in the genetics and ecology community to identify the genetic architecture of interspecies interactions and therefore more precisely predict the dynamic behavior of communities. (2) Genetic mapping or association studies is a routine approach for quantitative genetic research, but existing approaches are all based on the genotype-phenotype relationship of a single species. Given that the phenotype of one species is affected by other species in nature, these approaches without considering such interspecific interactions may provide misleading results about the inference of trait inheritance. Our framework is also among the first to precisely map complex traits by simultaneously estimating direct genetic effects, indirect genetic effects and genome-genome epistatic genetic effects of QTLs from different species. Since the second and third effects are largely omitted in current statistical genetics studies, our framework will provide results that can gain new insight into the comprehensive genetic architecture of complex phenotypes.

I have a few fundamental, major concerns with the manuscript that preclude my ability to recommend it for publication, along with many minor comments detailed as highlighted passages and comments in the attached Word file.

1) I am not convinced that the results are not simply the result of over-fitting, given that the authors make everything in the model time-dependent. It would be more rigorous to first fit a non-time-dependent model (fixed parameters, with direct and epistatic genetic effects on them) before introducing the enormous degrees of freedom granted by time-dependence.

Our response: We have quite a lot to say about this point. Yet, we would like to respect and adopt Dr. Friesen's suggestion because her comment always inspires our thinking about how to improve the presentation of our manuscript. Now, we have first modeled the data at individual time points and then compared the results from such a non-time-dependent model with those by the time-dependent model (see Fig. 2 of the current version).

We feel that the degree of freedom is not only a criterion for evaluating the robustness of a model, which is also heavily dependent on the amount of data information. In our case, the non-time-dependent model uses phenotypic data measured at only one time point at a time, whereas the time-dependent model capitalizes on all data measured at all 16 time points, 0, 2, 4, 6, 8, 10, 12, 14, 16, 18, 20, 22, 24, 28, 32, 36 hours, simultaneously. As such, the information used by the time-dependent model is larger by 15 times than that by the no-time-dependent model. On the other hand, using the non-time-dependent model, one needs to estimate two genotypic values and a residual variance, totaling to three unknown parameters, compared with one mean parameter and a residual variance under the null hypothesis (a total of two unknown parameters). Thus, the non-time-dependent model has $3 - 2 = 1$ degree of freedom. Because of the implementation of two-parameter Richards growth equation, the time-dependent model need to estimate $4 \times 2 = 8$

parameters for two QTL genotypes and two covariance-structuring parameters, with a total of 10 unknown parameters. Under the null hypothesis, a total of unknown parameters to be estimated is $4 + 2 = 6$. Therefore, the degree of freedom is $10 - 6 = 4$ for the time-dependent model.

Although the time-dependent model has three more degrees of freedom than the no-time-dependent model, the former has an amount of phenotypic information 15 times larger than the latter. We feel that the time-dependent model can be much more powerful and robust than the no-time-dependent model. Our data analysis has very well validated this argument.

Dr. Friesen suggested mapping parameters estimated by fitting each strain. This approach is OK, but it has proven not to be as good as mapping curves in a unifying framework. First, fitting each strain would lead to tremendous errors (you can imagine you are fitting a complicated curve only using one individual. In statistics, this would bring big errors). Second, mapping parameters as a phenotype will bring a second round of errors. This implies that mapping parameters will have double errors, especially with errors in the first stage being huge.

As a statistician, the senior author is quite familiar with the current status of statistical modeling and analysis of dynamic and longitudinal data. Parametric fitting of time-varying data, which has well been developed in the statistical literature, was used to handle our microbial growth data. We feel very confident about the statistical vigor and robustness of our approach used.

2) The relationship of the new method to previous methods is not well-explained. In a few places, comparisons were made between the new method and a previous method, but other than the new method having much smaller p-values (of which I am skeptical, due to comment 1), it is not clear what objective criteria were employed to compare the approaches.

Our response: The previous model is functional mapping that maps a single dynamic trait of a single species, whereas the new model maps ecologically related traits of two species that change over time. Functional mapping uses a single-trait growth equation, whereas the new model uses a system of ordinary differential equations.

Both models have been used to analyze the data, showing that the new model has greater power for QTL detection than functional mapping. To validate this finding, we have performed computer simulation studies by mimicking the real example. It turns out that the new model has greater power than the previous model when data contains interspecific interactions (please see Supplementary Table 2 and 3).

This issue was asked by Reviewer #1 and, therefore, has been addressed in the first revision. It seems that our previous reply has satisfied Reviewer #1. Please see his/her comment: *“In particular, the simulations and the additional text on the statistical methods have improved the paper.”*

3) The simulation portion of the study similarly lacked a rigorous presentation--how many loci contributed to the phenotype? What was the population structure of the simulated genomes? Were type I and type II error calculated and if so what were they as a function of sample size?

Our response: The objective of computer simulation is to investigate the statistical properties of the new model. These properties include the accuracy and precision of parameters (genotypic curves in our case here), power for QTL detection and false positive rates. We have mimicked the data structure of the example in terms of markers and population structure.

In statistical genetics, we usually use “power” rather than “type I error”, although these two are exchangeable. The expression of Type I error needs the theoretical derivation which relies on data distribution, whereas Power can be calculated empirically from simulation. Similarly, we often use false positive rates rather than “Type II error.” Power and false positive rates have been given in Supplementary Table 2 and 3, with results under different sample sizes.

4) It was not clear whether appropriate multiple-testing corrections were employed. These are relatively small numbers of genomes, but for loci of strong effect it is possible to obtain significant genotype-phenotype correlations after Bonferroni correction or FDR. Please clarify this important point--using a nominal p-value of 5% would be predicted to give more hits than observed, given that there are ~100K SNPs per genome.

Our response: We did not use multiple testing corrections. However, we have used permutation tests to determine the critical thresholds.

Bonferroni correction or FDR are often used by biologists, but in many cases they are quite misleading. For example, Bonferroni correction is often conservative. However, permutation tests that do not rely trait distribution and even sample size are more powerful and more appropriate to determine the thresholds. Statisticians incline to use permutation tests because they understand advantages and disadvantages of different statistical approaches. Permutation tests can determine the genome-wide critical threshold at any significance level (5%, 1% or 0.1%).

5) The conclusion that the genes identified were biologically meaningful based on the fact that they had annotation did not make sense to me. Was a formal GO-enrichment or other functional analysis conducted?

Our response: The functional annotation of all genes was made by submitting gene sequence data to NCBI's GenBank[®] sequence database. This is a very powerful database with detailed information about gene function. Gene annotation provides more detailed information than GO-enrichment analysis. Please see Supplementary Table 1 that details the name and function of each gene.

6) The relationship of the method and "game theory" was unclear. There did not seem to be any payoff matrices or fitness functions as in classical game theory or its modern descendent, adaptive dynamics. Furthermore, these short-term growth experiments would not have much opportunity for actual fitness, not was there competition between alternative alleles or phenotypes. Please clarify this connection or leave it out as it seems to promise something that is not delivered.

Our response: This is a very valuable comment. Again, we appreciate Dr. Friesen's thoughtful thinking and reading. We agree that the previous version has not well presented a relevance of our model to game theory. We have modified our model by linking it to community ecology. Community ecology is the theory of studying how species interact with each other to affect community dynamics. We have changed the title as "**A competition-cooperation mapping framework of community dynamics.**"

7) I believe there is a logical error in assuming that all pairs of species in fact are able to coexist with one another. Since this is never explicitly tested, all of the discussion about coexistence ought to be severely qualified or removed.

Our response: We respect and adopt Dr. Friesen's comment by removing coexistence.

8) In batch culture, Michaelis-Menton or Gompertz growth functions can sometimes fit better than logistic growth. These alternative forms should be explored, particularly since there do not seem to be any conceptual reasons for preferring the logistic form with interaction coefficients. In particular, if cross-feeding is occurring this could be modeled explicitly. However, if interference competition is present then perhaps the LV form would in fact be the best model. In any case, this should be assessed and justified.

Our response: This is a great point. We have compared several commonly used microbial growth equations including the Gompertz, logistic, Richards and LV. By the F test, we found the Richards equation outperforms the Gompertz and logistic equations in monoculture. The LV equations are the best for co-culture according to AIC.

We feel that the Michaelis-Menton equation describes how microbial growth responds to medium concentration, which is irrelevant to our study here.

9) Overall I felt that the manuscript could be better structured so that it was clear what exactly the authors did and what precise outcomes were obtained, rather than having everything in the methods it would help to have some framing in the main text. The claims also seemed to be overstated and not clearly justified by the results.

Our response: Once again, thank you, Dr. Friesen, for your tremendous effort and time given to our manuscript. As you can see, we have done our best to consider your comments and further incorporate each of them into this revision. In particular, we have re-structured the paper according to your comment which, we believe, has led to a much better presentation.

Even with these concerns, I do think that applying GWAS methods (which appears to be what they actually did, rather than QTL mapping?) to ecological data such as these is very interesting, though it does not seem to have any connection to game theory. I would strongly suggest taking a step back and starting with time-independent parameters as the phenotypes rather than the abundances at each time point.

Our response: Thank you for your constructive comment on the use of genetic approaches to study ecological problems. Generally speaking, QTL (linkage) mapping is used for controlled

crosses, whereas (linkage-disequilibrium-based) GWAS is based on multiple varieties sampled from a natural population. However, “mapping” can be used in both linkage mapping and GWAS. For example, some authors call association studies as association mapping. The detection of QTLs is the same goal of QTL mapping and GWAS. We tend to use “mapping” because our model can also be used for controlled crosses (not only natural populations).

We have switched our attention from game theory to community ecology. We believe community ecology is more relevant to what we are proposing (thanks are due to Dr. Friesen for her valuable point here).

As suggested, we have first analyzed no-time-dependent data, followed by a time-dependent data analysis.

Reviewers' comments:

Reviewer #4 (Remarks to the Author):

The authors present a framework that identifies traits relevant for ecological interactions by linking loci in strain genomes belonging to two different species to co-culture data via growth models.

Overall, this is an interesting and elegant study and I recommend its publication, once the authors have addressed my comments below.

The authors fail to discuss the assumptions and limitations of their method. One limitation is that pair-wise species dynamics may not always be captured by the standard growth models and may in some cases require kinetic parameters. For instance, Momeni & colleagues pointed out that the LV equation fails to describe some interaction mechanisms (<https://elifesciences.org/articles/25051>). The authors should point out the assumptions LV makes about the co-culture dynamics (additivity).

In this context, I have to mention another shortcoming: it is my understanding that each strain and each strain pair was cultivated only once, without biological replicates. While this is acceptable given the goals of this study, the authors could discuss how their method would deal with biological variation between replicates.

Although the authors give some information about significant QTLs, it would be interesting to dig a bit more into their interpretation. What (if anything) is known about the interactions between *S. aureus* and *E. coli*? Given the QTLs identified, what could be possible interaction mechanisms? At least, the authors could find out which pathways are affected.

In their answer to reviewer 3, who is concerned about the sensitivity (power, type I) and specificity (type II) of CCM, the authors refer to their supplementary tables. I have checked these, but did not find any data on specificity. The authors should compute the positive predictive value (PPV) of FunMap and CCM. PPV is defined as the ratio of true positives to all positives and thus avoids true negatives, which in cases where there are too many lead to bias. The power alone is uninformative when it comes to the assessment of a method's accuracy.

Finally, the authors argue in their rebuttal that QTLs identified through permutation tests do not need to be corrected for multiple testing. But in the end, if my understanding is correct, they test hypothesis (11) as often as there are trait combinations. In short, they test a hypothesis multiple times. Thus, a multiple testing correction is necessary.

Reviewer #1

Comments from Reviewer 1 on responses to Reviewer 3's points

I have looked again at the revised version of the manuscript "A competition-cooperation mapping framework of community dynamics" in light of the comments of referee 3. I think most of the major comments were addressed in a satisfactory way that, however, needs to be made more explicit in the manuscript itself.

In particular, the authors make a point that the analysis by means of a time-dependent model is more appropriate than a time-independent model (point 1). It is worth emphasizing in the text that the fit is not to a free time-dependent function of 16 time-points, but to an established model-based family of curves with 4 parameters (the Richardson equation, eq. 1c). The considerations on model complexity (reply to point 1) should appear in the manuscript itself. I would also like to see a Supplementary Table with the explicit posterior log likelihood values for the significant inferences, together with the posterior values of θ and the corresponding posterior values for the subleading models (including the time-independent model). Are all 4 parameters of the Richardson model needed or is a model with fewer parameters sufficient (say, constant shape parameter)? Can the complexity of the full model be justified by a Bayesian information criterion?

Similarly, the answer to point 4 of the referee (multiple testing correction) should be made completely transparent in the manuscript. What exactly was permuted in the statistical test? Maybe it should be mentioned that the test is more stringent than a Bonferroni correction (reply to point 4).

With these additions, I can recommend the paper for publication.

Reviewer #4

The authors present a framework that identifies traits relevant for ecological interactions by linking loci in strain genomes belonging to two different species to co-culture data via growth models.

Overall, this is an interesting and elegant study and I recommend its publication, once the authors have addressed my comments below.

Our response: Thank you for your constructive comments. We have very carefully read your comments and incorporated all of them into the manuscript. Below are the explanations about how we have addressed your comments.

The authors fail to discuss the assumptions and limitations of their method. One limitation is that pair-wise species dynamics may not always be captured by the standard growth models and may in some cases require kinetic parameters. For instance, Momeni & colleagues pointed out that the LV equation fails to describe some interaction mechanisms (<https://elifesciences.org/articles/25051>). The authors should point out the assumptions LV makes about the co-culture dynamics (additivity).

Our response: First of all, thank you for drawing this important article into our attention. The authors of this article showed that standard pairwise LV equations may fail to capture the complexity of microbial interactions and argued that a mechanistic model that implements chemical mediators of interactions is crucial for predicting multiple microbes in interactive communities. We agree with you that the limitation of the standard LV models should be discussed.

We have written a paragraph about this issue in the Discussion part (highlighted in red), which is shown here: "Third, additive Lotka-Volterra pairwise models can only characterize how one species stimulates or inhibits the abundance of other species in a gross way. By considering the chemical mediators underlying interspecific interactions, Momeni et al. theoretically showed that these additive models fail to capture the complexity of ecological interactions⁴¹. These authors have formulated mechanistic reference models for predicting two different species engaging in chemical-mediated interactions. By incorporating Momeni et al.'s mechanistic models, CoCoM can be armed to establish a more precise predictive model of community dynamics and evolution of interacting species in ecological systems."

In this context, I have to mention another shortcoming: it is my understanding that each strain and each strain pair was cultivated only once, without biological replicates. While this is acceptable given the goals of this study, the authors could discuss how their method would deal with biological variation between replicates.

Our response: We have performed three replicates for both monoculture and co-culture, but we are sorry for not making this clear in the previous version of the manuscript. In the Study material subsection of this version, we have given a couple of sentences to show this replication (highlighted in red).

Although the authors give some information about significant QTLs, it would be interesting to dig a bit more into their interpretation. What (if anything) is known about the interactions between *S. aureus* and *E. coli*? Given the QTLs identified, what could be possible interaction mechanisms? At least, the authors could find out which pathways are affected.

Our response: CoCoM can chart a more complete picture of genetic architecture by revealing

previously neglected indirect and genome-genome epistatic genetic effects of QTLs. We explained this power based on a representative QTL pair E4614704 (from *E. coli*) and S188004 (from *S. aureus*). E4614704 resides in *yjjW*, a gene that encodes a homolog of pyruvate-formate lyase activating enzyme PflA (Kolker et al. 2004). As a key intersection in the network of metabolic pathways, pyruvate can be converted to carbohydrates via gluconeogenesis, fatty acids or energy through acetyl-CoA, the amino acid alanine, or ethanol, depending on whether the condition is aerobic or anaerobic. All these metabolic processes may play an important role in adapting *E. coli* to microbial coexistence. S188004 is relevant to the gene *ggt* (Wu et al. 2004). This gene encodes gamma-glutamyltranspeptidase that regulates the metabolic pathway of glutathione. By converting methylglyoxal to D-lactate, glutathione may form a key pathway for *S. aureus* to react with microbial interactions.

In their answer to reviewer 3, who is concerned about the sensitivity (power, type I) and specificity (type II) of CCM, the authors refer to their supplementary tables. I have checked these, but did not find any data on specificity. The authors should compute the positive predictive value (PPV) of FunMap and CCM. PPV is defined as the ratio of true positives to all positives and thus avoids true negatives, which in cases where there are too many lead to bias. The power alone is uninformative when it comes to the assessment of a method's accuracy.

Our response: We have followed the reviewer's comment by computing false positive rates (specificity). Thus, we have given both sensitivity and specificity of the new model (in the supplementary tables).

Finally, the authors argue in their rebuttal that QTLs identified through permutation tests do not need to be corrected for multiple testing. But in the end, if my understanding is correct, they test hypothesis (11) as often as there are trait combinations. In short, they test a hypothesis multiple times. Thus, a multiple testing correction is necessary.

Our response: This is a statistical issue. Permutation tests have been routinely used as an empirical approach for determining the critical threshold of QTL detection in mapping and association studies. This approach is such popular that we don't need to cite the original reference (Churchill and Doerge 1994). By re-sampling phenotypic data from the original genotype and phenotype data pool, the cut-off point from permutation tests presents a genome-wide significance level, and therefore, no multiple comparison is needed. The corresponding author is a Professor of statistics and has published extensively permutation test papers in statistical and genetic journals. We believe that we have very well addressed this issue.

Churchill, G. A., and R. W. Doerge, 1994 Empirical threshold values for quantitative trait mapping. *Genetics* 138: 963–971.

Reviewer #1

I have looked again at the revised version of the manuscript "A competition-cooperation mapping framework of community dynamics" in light of the comments of referee 3. I think most of the major comments were addressed in a satisfactory way that, however, needs to be made more explicit in the manuscript itself.

In particular, the authors make a point that the analysis by means of a time-dependent model is more appropriate than a time-independent model (point 1). It is worth emphasizing in the text that the fit is not to a free time-dependent function of 16 time-points, but to an established model-based family of

curves with 4 parameters (the Richardson equation, eq. 1c). The considerations on model complexity (reply to point 1) should appear in the manuscript itself.

Our response: If we understand your question correctly, you wanted us to show in the text that we fit the time-series data to growth equations? Yes, we did so (highlighted in red, page 5-6). Below are two specific descriptions in the text:

"By assuming the four-parameter Richards equation as one that can exactly predict microbial growth, we implemented a statistical procedure to test and validate whether any of the three-parameter Gompertz and logistic equations can sufficiently describe the data. Results from the *F* test suggest that the Richards equation provides an optimal fitness to time-dependent abundance data for both *E. coli* and *S. aureus* in monoculture (Fig. 1)."

"We fit the growth data of the two bacterial species in co-culture using Gompertz, logistic, Richards, and LV equations (Supplementary Table 1) and further chose one that best fit the data based on the AIC information criterion. The result suggests that the LV equations outperform the Gompertz, logistic and Richards equations in fitting growth trajectories of both species in co-culture (Fig. 1)."

Note: Figure 1 shows what optimal growth equation has been selected for monoculture and co-culture.

I would also like to see a Supplementary Table with the explicit posterior log likelihood values for the significant inferences, together with the posterior values of θ and the corresponding posterior values for the subleading models (including the time-independent model). Are all 4 parameters of the Richardson model needed or is a model with fewer parameters sufficient (say, constant shape parameter)?

If we understand your question correctly, you wanted us to tabulate growth parameter estimates by a Bayesian approach? We used the likelihood approach, but did not use the Bayesian approach. The estimation of growth equations is a standard statistical approach reported widely in the literature. However, we have still adopted your suggestion by giving a supplementary table to list the estimates of growth parameters in both monoculture and co-culture.

Note: The choice of the best equation was based on F-test for monoculture and AIC for co-culture. This has been made clear in the text. In monoculture, the Richards was chosen and in co-culture, LV equations were chosen.

Can the complexity of the full model be justified by a Bayesian information criterion?

If we understand your question correctly, we wanted us to show how the model was chosen? We reported the result from AIC, but we did not use BIC.

Similarly, the answer to point 4 of the referee (multiple testing correction) should be made completely transparent in the manuscript. What exactly was permuted in the statistical test? Maybe it should be mentioned that the test is more stringent than a Bonferroni correction (reply to point 4).

Our response: This is a statistical issue. Permutation tests have been routinely used as an empirical approach for determining the critical threshold of QTL detection in mapping and association studies. This approach is such popular that we don't need to cite the original reference (Churchill and Doerge 1994). By re-sampling phenotypic data from the original genotype and phenotypic data pool, the

cut-off point from permutation tests presents a genome-wide significance level, and therefore, no multiple comparison is needed. The corresponding author is a Professor of statistics and has published extensively permutation test papers in statistical and genetic journals. We believe that we have very well addressed this issue.

Churchill, G. A., and R. W. Doerge, 1994 Empirical threshold values for quantitative trait mapping. *Genetics* 138: 963–971.

With these additions, I can recommend the paper for publication.

Our response: Thank you very much for your time and effort given to our manuscript. Your previous and current comments have been very helpful for us to revise and improve the manuscript.

REVIEWERS' COMMENTS:

Reviewer #1 (Remarks to the Author):

I have looked at the second revised version and the response to reviewers' questions. I am happy to now recommend the ms. for publication. There are no more comments to address.

Reviewer #4 (Remarks to the Author):

The authors have addressed almost all of my comments. The only missing analysis is the pathway mapping, where enzyme-coding genes with significant QTLs are highlighted on pathway maps. Since it is straightforward to carry out this analysis (e.g. with the KEGG pathway mapper: http://www.genome.jp/kegg/tool/map_pathway2.html) and to add a summary of the results to the supplement, I now recommend the work for publication.

**Reviewer #4**

The authors have addressed almost all of my comments. The only missing analysis is the pathway mapping, where enzyme-coding genes with significant QTLs are highlighted on pathway maps. Since it is straightforward to carry out this analysis (e.g. with the KEGG pathway mapper: http://www.genome.jp/kegg/tool/map_pathway2.html) and to add a summary of the results to the supplement, I now recommend the work for publication.

**Our response:** Thank you very much for your time and effort given to our manuscript.

We have made an in-depth analysis of a representative pair of QTLs detected from two species The pathway map for gene *ggt* (nearby QTL S188004 detected in *S. aureus* by CoCoM) is given in Supplementary Fig. 2 of this version (please see below). *ggt* is the gene encoding gamma-glutamyltranspeptidase (EC 2.3.2.2).

The gene yjjW (detected from *E.coli*) encodes a putative pyruvate formate lyase activating enzyme. There is no KEGG map identified for it.